# RL on Incorrect Synthetic Data Scales the Efficiency of LLM Math Reasoning by Eight-Fold

**Amrith Setlur**[*,1] **Saurabh Garg**[1] **Xinyang (Young) Geng**[2] **Naman Garg**[3]
**Virginia Smith**[1] **Aviral Kumar**[1,2]
[1]Carnegie Mellon University    [2]Google DeepMind    [3]MultiOn

## Abstract

Training on model-generated synthetic data is a promising approach for finetuning LLMs, but it remains unclear when it helps or hurts. In this paper, we investigate this question for math reasoning via an empirical study, followed by building a conceptual understanding of our observations. First, we find that while the typical approach of finetuning a model on synthetic correct or *positive* problem-solution pairs generated by capable models offers modest performance gains, sampling more correct solutions from the finetuned learner itself followed by subsequent fine-tuning on this self-generated data **doubles** the efficiency of the same synthetic problems. At the same time, training on model-generated positives can amplify various spurious correlations, resulting in flat or even inverse scaling trends as the amount of data increases. Surprisingly, we find that several of these issues can be addressed if we also utilize *negative* responses, *i.e.*, model-generated responses that are deemed incorrect by a final answer verifier. Crucially, these negatives must be constructed such that the training can appropriately recover the utility or advantage of each intermediate step in the negative response. With this *per-step* scheme, we are able to attain consistent gains over only positive data, attaining performance similar to amplifying the amount of synthetic data by **8×**. We show that training on per-step negatives can help to unlearn spurious correlations in the positive data, and is equivalent to advantage-weighted reinforcement learning (RL), implying that it inherits robustness benefits of RL over imitating positive data alone.

## 1 Introduction

Training large language models (LLMs) relies on the ability to train on large amounts of high-quality data. It is predicted that we will run out of high-quality internet data by 2026 [32, 58], necessitating training on model-generated data, or what is commonly referred to as *synthetic data*. Recent trends illustrate that scaling up synthetic data can lead to improvements [8, 29] on hard reasoning problems, while other results illustrate that training on synthetic data can steer the performance of the model into a downward spiral [3, 17, 51]—amplying biases, misinformation, and undesired stylistic properties. Thus while *in principle*, synthetic data could potentially address data scarcity, it must be designed in an appropriate manner to be effective. However, this has been hard due to a lack of an understanding of how synthetic data contributes to LLM behavior.

To provide clarity on how synthetic data contributes to performance, we aim to understand its impact on LLM capabilities via a study on math reasoning, a prevalent scenario where synthetic data is used. Typically, in this setting, synthetic data corresponds to correct or *positive* model-generated responses for a novel set of initial problems synthesized by prompting capable models [29, 31]. The resulting model is then evaluated on a held-out set of problems drawn from a test set. Perhaps as expected, we find that performance improves when finetuning models on positive synthetic responses, though the scaling rates for performance improvement are often substantially slower than those observed during pretraining. Concretely, we find that under the scaling law of Zhang et al. [71], the error rate scales as $\approx D^{-0.05}$ to $D^{-0.15}$ in the size $D$ of synthetic dataset. Second, we observe that not all types

---

[*]*Corresponding author. Please send correspondences to asetlur@andrew.cmu.edu.*

38th Conference on Neural Information Processing Systems (NeurIPS 2024).

of positive synthetic data are equally effective: often positive responses self-generated by the learner itself are as effective as 2× synthetic data from bigger models in improving performance. This is because responses from a similar model are "easier-to-fit" than those from a more capable model, resulting in reduced memorization [26, 56] during finetuning. We also observe that if the positive response contains incorrect/irrelevant intermediate steps, training on such data often incentivizes the model to overfit on spurious correlations, leading to a flat or even inverse scaling with more data.

Perhaps surprisingly, we find that the aforementioned pathologies of training on positive data only can be addressed if we also utilize synthetic *negative* responses: responses generated by the model that do not result in obtaining a correct final answer. One way to utilize negative responses is via methods such as direct preference optimization (DPO) [41]. While performance of standard DPO [41] largely flatlines as the number of synthetic problems are scaled up (Figure 5), we are able

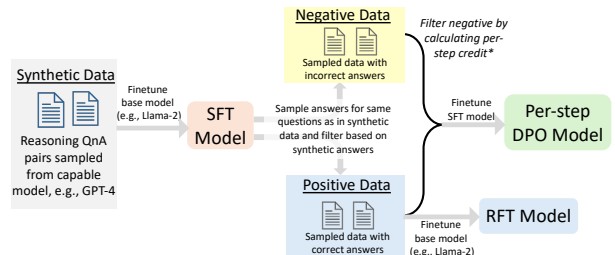

**Figure 1:** *Positive and negative synthetic data:* Pictorial representation of positive/negative synthetic data definitions we use and how they are fed to SFT, RFT and DPO.

to attain consistent improvements if the negative data is generated appropriately. A solution trace for a math problem typically comprises of multiple *reasoning steps* corresponding to intermediate results. Our insight is that instead of contrasting arbitrary correct and incorrect responses, we should contrast those positive and negative responses that depict good and bad choices for the more "critical" intermediate steps: steps that the model must carefully produce so as to succeed at the problem. In other words, critical steps are those which the model is unable to recover from, and hence, must be emphasized. With this scheme, we are able to attain consistent gains over only positive data, **attaining performance similar to scaling up positive synthetic data by 8×.** We show that training on this sort of negative data evades spurious steps amplified by training on positive data alone.

To theoretically understand our findings, we build a conceptual model of how training on this data benefits performance. Formally, we show that this construction of negative data, which emphasizes "critical" tokens (Figure 6) enables us to perform credit assignment, and is equivalent to training the model with per-step advantage-weighted reinforcement learning (RL) [40] on a mixture of positive and negative synthetic data. Specifically, these advantage values are computed under an optimal value function induced by sampling multiple responses under the SFT policy obtained by training on only the positive data. This reduction of using negative data to advantage-weighted RL enables us to conceptually compare it to training on positive data, which corresponds to imitation learning (*i.e.*, behavioral cloning) on positive data. First, we are able to argue for the generalization gains of advantage-weighted RL through the lens of distribution robust objectives. Second, building on theoretical results in RL [27], we are also able to show that when advantages can be estimated reliably, advantage-weighted RL will be significantly more sample-efficient compared to imitation. Overall, this model explains the utility of negative data over only positive data.

Our contribution is a study of the role of synthetic data in improving math reasoning capabilities of LLMs. We derive scaling laws for positive and negative data on common reasoning benchmarks and observe that: **(a)** training on positive synthetic data from capable models results in scaling rates that are significantly slower than standard empirical risk minimization; **(b)** training on model-generated positive synthetic data can improve sample efficiency by 2× but also amplifies spurious correlations; **(c)** appropriate ways of constructing learner-specific negative data with emphasis on critical steps, results in a performance boost equivalent to scaling up positive data 8×; **(d)** training with negative data provides a mechanism to unlearn spurious correlations; and **(e)** we present a conceptual model inspired from RL to explain our observations on synthetic data and the generalization benefits we see.

## 2   Related Work

A standard procedure to finetune a pretrained LLM is teacher-forcing on expert data, *i.e.*, maximizing the likelihood of the next token given all previous tokens [7, 61]. In Appendix G we discuss some failure modes of this procedure for math reasoning that positive or negative synthetic data can address.

**Positive synthetic data.** Learning theory dictates that the SFT policy trained on more SFT data (*e.g.*, 1.5M for DeepSeek-Math [5]) would have improved math reasoning capbabilities. Thus, a

common goal for generating synthetic data as close as possible to the SFT data [29, 31, 32]. That said, generating high quality math data can be challenging, since verification can often be hard. When synthetic data is verified by larger models [50, 59], recent works [33, 66] observe scaling similar to finetuning LLMs on expert data [69, 71], while another work [14] notes the compositional gains from SFT data for code generation. Common sources of "good" synthetic data include responses from stronger teachers [29, 30], or data generated by the SFT policy itself, in the framework of reinforced self-training (ReST) and STaR [8, 52, 69, 70]. In our work, we study and compare the performance scaling with positive synthetic data from bigger models like GPT-4 and Gemini 1.5 Pro with self-generated positive data. We connect our findings to evidence showing "ease of learning" generalizable features on self-generated completions [26] which often prevents undesirable memorization [56]. Finally, our work also sheds light on several concerns about training on synthetic positive data amplifying biases [48, 63], and leading to model collapse [13, 17], especially due to overfitting on"spurious" intermediate steps. We conceptually explain this phenomenon and also discuss how negative model-generated responses can help identify and unlearn those spurious steps.

**Benefits and nuances of negative synthetic data.** While most works on synthetic data [29, 32, 66, 69] train only on correct answers, our work also studies complementary gains from incorrect completions generated by the SFT policy [23, 38, 39, 68]. To leverage sub-optimal negative data, we adopt the framework of offline preference optimization [16, 41, 73], where a preference pair is constructed using correct and incorrect responses for the same problem [38]. Despite numerous studies on preference data composition [8–10, 37, 54, 55, 60], its unclear how to pose a reasoning problem as a preference optimization problem. Randomly pairing correct and incorrect completions in a preference pair can lead to poor performance [21, 38, 39, 64] due to objective mismatch [55, 72] and requires auxilliary losses to perform well. Another option is to use negative data for training verifiers [22, 65] but this line of work still only trains the policy using positive data. We introduce a conceptual model of negative data, where we understand how certain choices of negative data can assign per-step credits, which we use to establish the equivalence of preference optimization to advantage weighted RL. Self-explore method in Hwang et al. [23] can be viewed as an special instance of our general framework. Other works [34, 59] exploit per-step credit assignment through tree-based sampling. They identify the reasoning subsequence that led to the most incorrect answers under the SFT policy for training a reward model. While this is related, our conceptual model and analysis also understands why assigning per-step credits can generalize better by unlearning spurious correlations, *e.g.,* when the credits are given by the Q-function of the "best-of-K" SFT policy.

## 3    Problem Setup and Synthetic Data Generation Pipeline

Building on the recipe of Li et al. [29], Liu et al. [31], we use GSM8K [11] and MATH [19] to collect synthetic data consisting of both novel problems designed by capable models such as GPT4 [1] and Gemini 1.5 Pro [44], and responses to these problems, obtained from the same models.

**Synthetic data pipeline.** First, given a dataset $\mathcal{D}_{\text{real}} = \{(x_i^r, y_i^r)\}$ of problems $x_i^r \sim p_{\text{real}}(x)$ and solution traces $y_i^r \sim p_{\text{real}}(y \mid x_i)$, we prompt one of the highly-capable models with a uniformly random sample $(x_i^r, y_i^r) \in \mathcal{D}_{\text{real}}$ and ask the model to generate a new problem $x_i$ such that it is similar to the real problem $x_i^r$, in a way that a feasible solution exists. Second, we ask the model to provide a solution trace answer $y_i$ with step-by-step reasoning (exact prompts for $x_i, y_i$ are borrowed from Li et al. [29], shown in Appendix H). We assume that the answers generated via this process are accurate, and perform lightweight filtering step to remove duplicates, badly-formatted answer traces, and model failures. Based on the above, for any synthetic problem and solution pair $(x, y)$, we can define a binary reward function $r(y, \hat{y}) \mapsto \{0, 1\}$, which verifies if a new solution trace $\hat{y}$ is correct or not. This is implemented with a set of answer extraction and string matching tools borrowed from [29, 66]. We say that a new trace $\hat{y}$ is a *positive* trace if it produces the correct final answer *i.e.,* $r(\hat{y}, y) = 1$, and *negative* if it produces an incorrect final answer, *i.e.,* $r(\hat{y}, y) = 0$. By definition, $r(y, y) = 1$, and the original trace $y$ is always positive.

**Positive and negative datasets.** The above process induces a joint distribution $p_{\text{syn}}(x, y)$, *iid* samples from which yields positive synthetic dataset $\mathcal{D}_{\text{syn}}$. We note that the sampling process for $\mathcal{D}_{\text{syn}}$ is designed to ensure that the induced marginal distribution over synthetic problems $p_{\text{syn}}(x)$ is close to $p_{\text{real}}(x)$. We will use $\mathcal{D}_\pi^+$ to denote the positive dataset of $(x, +\hat{y})$ where $+\hat{y}$ is a positive solution trace generated from some policy $\pi(\cdot \mid x)$. For a positive $+\hat{y}$ and negative $-\hat{y}$ trace, sampled from the same policy $\pi(\cdot \mid x)$, we denote a dataset over problems and solution pairs: $(x, +\hat{y}, -\hat{y})$ as $\mathcal{D}_\pi^\pm$.

**Reasoning steps.** The trace $\boldsymbol{y}_i$ consists of several intermediate steps, $\boldsymbol{y}_i = \left[ \boldsymbol{y}_{i,1}, \ldots, \boldsymbol{y}_{i,L} \right]$. We assume each trace has at most $L$ steps, and use $\boldsymbol{y}_{1:t}$ to denote the subsequence of first $t$ steps. Since mathematical reasoning problems require step-by-step computation, simply arriving at an incorrect final answer does not mean that all steps in a negative $\hat{\boldsymbol{y}}$ are incorrect. Similarly, a positive $\hat{\boldsymbol{y}}$ may also have incorrect reasoning steps. In fact, even the original answers generated by more capable models in $\mathcal{D}_{\mathrm{syn}}$ may also contain incorrect reasoning steps, and training on such traces may actually lead to unintended consequences (Section 5).

## 4 Learning from Synthetic Data

In this section, we discuss various algorithms for learning from the synthetic dataset $\mathcal{D}_{\mathrm{syn}}$ discussed in the previous section, as well as positive and negative solution traces generated using a model.

**Supervised and rejection finetuning (SFT and RFT).** Given positive synthetic $\mathcal{D}_{\mathrm{syn}}$, perhaps the most straightforward approach (and the most prevalent) is to learn $\pi_{\mathrm{sft}}$ on this data via supervised next-token prediction: $\pi_{\mathrm{sft}}(\cdot|\boldsymbol{x}) := \arg\max_{\pi} \mathbb{E}_{\boldsymbol{x},\boldsymbol{y}\sim\mathcal{D}_{\mathrm{syn}}} \left[ \log \pi(\boldsymbol{y}|\boldsymbol{x}) \right]$. Another option is to train via supervised next-token prediction on problems in $\mathcal{D}_{\mathrm{syn}}$, but when using a positive solution trace $\hat{\boldsymbol{y}}$ sampled from $\pi_{\mathrm{sft}}(\cdot|\boldsymbol{x})$, instead of positive synthetic responses from the capable models in $\mathcal{D}_{\mathrm{syn}}$. Akin to rejection finetuning (RFT [69] or STaR [70]), sampling from $\pi_{\mathrm{sft}}(\cdot \mid \boldsymbol{x})$ once is not guaranteed to give a positive response, and we instead sample $M$ times for each $\boldsymbol{x}$ and construct the dataset $\mathcal{D}^{+}_{\pi_{\mathrm{sft}}}$ of SFT policy generated positive responses. Then, we apply the next-token prediction loss on $\mathcal{D}^{+}_{\pi_{\mathrm{sft}}}$.

**Preference optimization.** Beyond positive data, we can also learn from negative synthetic data generated from the SFT policy, especially when contrasted with positive responses. However, learning from negative data presents multiple open design questions pertaining to the construction of negative traces, and the choice of the loss function, and simple supervised fine-tuning will not be a good choice since it will incentivize the model to produce more errors. Therefore, we use a contrastive training approach, direct preference optimization (DPO [41]) for incorporating negative data from $\pi_{\mathrm{sft}}$. In a nutshell, DPO trains a policy using the following preference optimization objective:

$$\min_{\pi} \quad \mathcal{L}_{\mathrm{DPO}}(\pi) := \mathbb{E}_{(\boldsymbol{x},+\boldsymbol{y},-\boldsymbol{y})\sim\mathcal{D}^{\pm}_{\pi_{\mathrm{sft}}}} \left[ \sigma\left( \beta \log \frac{\pi(+\boldsymbol{y} \mid \boldsymbol{x})}{\pi_{\mathrm{sft}}(+\boldsymbol{y} \mid \boldsymbol{x})} - \beta \log \frac{\pi(-\boldsymbol{y} \mid \boldsymbol{x})}{\pi_{\mathrm{sft}}(-\boldsymbol{y} \mid \boldsymbol{x})} \right) \right]. \quad (1)$$

We consider two objectives that construct negative data and subsequently optimize Equation 1. The first variant is ***standard DPO*** [41], which samples negative data $-\hat{\boldsymbol{y}}$ from the $\pi_{\mathrm{sft}}$ (with rejection sampling) and adds $(\boldsymbol{x}, \boldsymbol{y}, -\hat{\boldsymbol{y}})$ to $\mathcal{D}^{\pm}_{\pi_{\mathrm{sft}}}$. The second variant is ***per-step DPO*** [23], which first samples a complete solution trace $\hat{\boldsymbol{y}}_{1:L}$ from $\pi_{\mathrm{sft}}$ and then determines the "first pit" $\hat{\boldsymbol{y}}_c$. The first pit $\hat{\boldsymbol{y}}_c$ is the step where any completion following the step: $\hat{\boldsymbol{y}}_{c+1:L} \sim \pi_{\mathrm{sft}}(\cdot \mid \boldsymbol{x}, \hat{\boldsymbol{y}}_{1:c})$ leads to incorrect answers in expectation under $\pi_{\mathrm{sft}}$. The triplet $(\boldsymbol{x}, \boldsymbol{y}, \hat{\boldsymbol{y}}_{1:c})$ is added to the preference dataset $\mathcal{D}^{\pm}_{\pi_{\mathrm{sft}}}$.

## 5 Positive Data Improves Coverage, But Amplifies Spurious Correlations

We first analyze the influence of scaling up positive synthetic data on GSM8K and MATH. In this experiment, we fine-tune DeepSeek-Math-7B [5] and LLama2-7B [57] models (details in Appendix J) on varying sizes of $\mathcal{D}_{\mathrm{syn}}$, constructed out of a 5:1 mixture of GPT-4-turbo [1] and Gemini-1.5 Pro [44]. We obtain a series of SFT policies on this data scaling ladder. We then train a series of models by running one iteration of RFT on data obtained from the SFT policies at each step.

**Scaling results with positive synthetic data GPT-4 and Gemini 1.5 Pro.** Since we assume that the more capable models generate correct solutions for new problems, by scaling $\mathcal{D}_{\mathrm{syn}}$ we are increasing *coverage* under $p_{\mathrm{real}}$, *i.e.*, adding new $\boldsymbol{x}, \boldsymbol{y}$ with non-zero probability under $p_{\mathrm{real}}$. In Figures 2(a,b), we plot the test error rate of the SFT policy as $\mathcal{D}_{\mathrm{syn}}$ is scaled. As expected, we observe that the test error rate on both GSM8K and MATH improves with more positive data. Further, by simply fitting the parametric scaling law from [71], for $D := |\mathcal{D}_{\mathrm{syn}}|$, we find that the scaling trends decay as $\approx D^{-0.15}$ on GSM8K and $\approx D^{-0.05}$ on the harder MATH dataset, with similar trends for the corresponding pass@5 error rates. Since these scaling trends are much more underwhelming than those for pre-training [20], this perhaps implies that samples in $\mathcal{D}_{\mathrm{syn}}$ are indeed improving coverage over samples in $p_{\mathrm{real}}(\boldsymbol{x}, \boldsymbol{y})$, but maybe not as efficiently as sampling *iid* samples directly from it.

**Scaling results with positive synthetic data from 7B SFT policy.** Previously, we scaled problems in $\mathcal{D}_{\mathrm{syn}}$ by querying GPT-4 and Gemini-1.5. Now, for existing problems in $\mathcal{D}_{\mathrm{syn}}$ we generate new responses by sampling from the $\pi_{\mathrm{sft}}$ trained on problems+solutions in $\mathcal{D}_{\mathrm{syn}}$. For any $(\boldsymbol{x}, \boldsymbol{y}) \in \mathcal{D}_{\mathrm{syn}}$

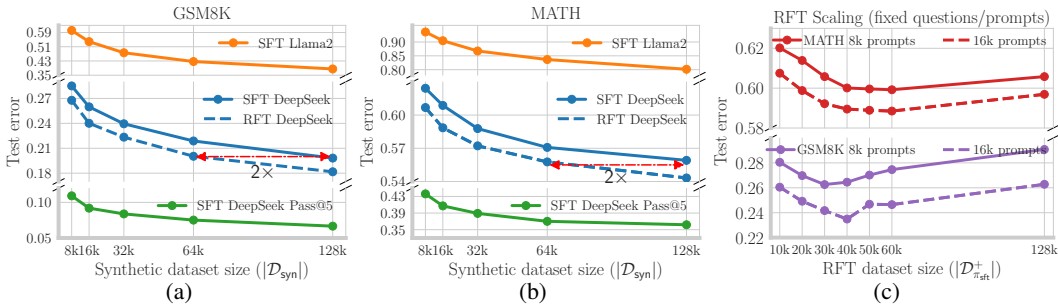

**Figure 2:** *Positive data scaling laws:* On GSM8K (a) and MATH (b), we evaluate SFT trained on $\mathcal{D}_{\text{syn}}$ and RFT that uses SFT policy generated positives ($\mathcal{D}^+_{\pi_{\text{sft}}}$), as we scale $\mathcal{D}_{\text{syn}}$, observing $\mathcal{D}^+_{\pi_{\text{sft}}}$ to be 2× as effective as $\mathcal{D}_{\text{syn}}$. In (c), we plot performance of RFT the number of correct solutions in $\mathcal{D}^+_{\pi_{\text{sft}}}$ are scaled, for a fixed set of 8k/16k problems from $\mathcal{D}_{\text{syn}}$, observing that scaling model positives can amplify spurious correlations.

we generate verified positive solution traces $\hat{y} \sim \pi_{\text{sft}}$ s.t. $r(\hat{y}, y) = 1$. Following Yuan et al. [67], to ensure we sample enough correct responses, we sample 100 times from $\pi_{\text{sft}}$ and generate RFT datasets $\mathcal{D}^+_{\pi_{\text{sft}}}$, where each problem has atmost 4 correct and diverse solutions. Next, we finetune the pretrained DeepSeek-Math-7B model on these new series of RFT datasets and plot the performance on GSM8K and MATH (Figure 2(a,b)). First, **we observe that for any size of $\mathcal{D}_{\text{syn}}$, the performance of the RFT model is better than the corresponding SFT model**, and the difference remains consistent as we scale $\mathcal{D}_{\text{syn}}$. Surprisingly, this indicates that training on positive answer traces from the 7B $\pi_{\text{sft}}(y \mid x)$ can lead to better performing policies than capable models.

**What is the value of positives from $\pi_{\text{sft}}(y \mid x)$?** If sampling from $\pi_{\text{sft}}$ also improves coverage and performance, then should we scale problems and solutions in $\mathcal{D}_{\text{syn}}$, or just solutions in $\mathcal{D}^+_{\pi_{\text{sft}}}$? To answer this, we need to assign a value to the RFT dataset $\mathcal{D}^+_{\pi_{\text{sft}}}$ in terms of $|\mathcal{D}_{\text{syn}}|$. We do this by training SFT policies on $\mathcal{D}_{\text{syn}}$ of sizes 8k and 16k, and then generating RFT datasets from the corresponding SFT policies where we only add more correct solution traces (for the same problems) and scale RFT data from 10k to 128k (unlike RFT data in Figure 2(a,b) where both questions and answers scale). In Figure 2(c) we plot the error rate of DeepSeek-Math-7B finetuned on the different sizes of $\mathcal{D}^+_{\pi_{\text{sft}}}$. Comparing the lowest values of the curves in Figure 2(c) with $\mathcal{D}_{\text{syn}}$ scaling in Figure 2(a,b), we note that **performance from $\mathcal{D}^+_{\pi_{\text{sft}}}$ is 2× the size of $\mathcal{D}_{\text{syn}}$ used to train $\pi_{\text{sft}}$.** We also note that performance can plateau (or worsen in the case of GSM8K) as we scale up $\mathcal{D}^+_{\pi_{\text{sft}}}$ by a lot. This is because $r(\cdot, y)$ is unable to verify the correctness of each step in the positive solution traces in $\mathcal{D}^+_{\pi_{\text{sft}}}$. Later, we see how incorrect steps induce spurious correlations that get amplified as we scale positive data, explaining this drop. See Appendix C for more discussion.

**Why is self-generated positive data more sample-efficient?**
From our result above, we find that solutions sampled from $\pi_{\text{sft}}$ (trained on $\mathcal{D}_{\text{syn}}$) yield better models, as good as those trained on $2 \times |\mathcal{D}_{\text{syn}}|$. This finding is surprising since one might expect more capable GPT-4/Gemini models to present better solutions, training on which should lead to good performance, akin to distillation [50], but this is not the case. Our results are consistent with the study of memorization in LLMs [18, 26, 56], which shows that pretrained (base) LLMs tend to memorize "hard-to-fit" and "out-of-pretraining-distribution" responses during finetuning, resulting in imperfect generalization. In contrast, correct response traces produced by $\pi_{\text{sft}}$ on problems from $\mathcal{D}_{\text{syn}}$

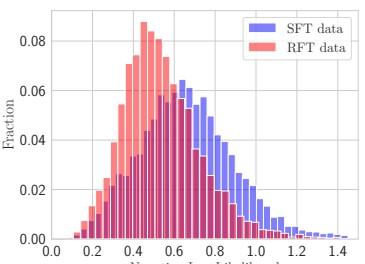

**Figure 3:** Under base LLM, $\mathcal{D}^+_{\pi_{\text{sft}}}$ has higher likelihood than $\mathcal{D}_{\text{syn}}$.

are not as hard-to-fit or as out-of-distribution, since they are obtained from a model that is "close" to the base LLM. We confirm this hypothesis with a histogram of negative log-likelihood values of the SFT and RFT data under the base LLM (Figure 3). Hence, we expect STaR/RFT to alleviate the memorization problem on a large chunk of examples. This finding also corroborates Yuan et al. [69]'s result that lower the perplexity of SFT data under the base model, the smaller the gap between SFT and RFT performance. Note that one may also attribute better performance of RFT to improved coverage from multiple answers in $\mathcal{D}^+_{\pi_{\text{sft}}}$ for each question in $\mathcal{D}_{\text{syn}}$. But, we find that even when RFT data is restricted to one solution per question, LLM trained on it outperforms SFT consistently by $> 1\%$. Since verification is cheap, we can sample more solutions and also benefit from coverage.

**SFT/RFT policy suffers from spurious correlations in positive synthetic data.** While RFT data maybe "easier-to-fit", in Figure 2(c) we also note that continuing to scale RFT data leads to test error saturation, or even worse test error. This is unlike scaling of problems and solutions in SFT data (in Figure 2(a,b)). This failure can be attributed to the presence of incorrect/irrelevant steps that are not detected by our verifier, since it only verifies the final answer (see Appendix J, K for examples). For a problem $x$, when the LLM is trained with supervised next-token prediction on some positive sub-optimal $y$ in the RFT data, with incorrect step $y_k$, it is likely to overfit on spurious correlations

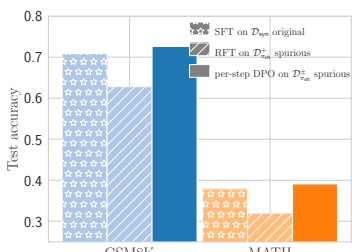

**Figure 4:** Spurious correlations in RFT data hurt performance.

between the sub-optimal subsequence $y_{1:k}$, and the following valid step $y_{k+1}$, when trying to maximize $\pi(y_{k+1}|y_{1:k}, x)$. To verify this hypothesis, we amplify the presence of these spurious steps. Specifically, for each question in $\mathcal{D}_{\mathrm{syn}}$ we sample "spurious steps" from $\pi_{\mathrm{sft}}$ trained on it, *i.e.*, steps which lead to the incorrect answer with high probability under $\pi_{\mathrm{sft}}$ (we sample multiple completions conditioned on the same spurious step to check how likely it leads to the correct final answer). Then, we interleave the solution traces in the RFT data with these spurious steps. Note, that all traces in the RFT data are still positive since, they all lead to the correct answer eventually. We find that the LLM trained on this sub-optimal spurious RFT data performs worse than the $\pi_{\mathrm{sft}}$ policy itself.

> **Takeaways for scaling positive synthetic data**
>
> - While positive data from GPT-4/Gemini-1.5 improves coverage over new problems and solutions, positive data from SFT policy trained on it is 2× more sample efficient.
> - Scaling positive data ($\sim \pi_{\mathrm{sft}}$) that contains spurious steps, leads to worse test errors.

## 6 Negative Synthetic Data Enables Per-Step Credit Assignment

The spurious correlations from Section 5 correspond to intermediate irrelevant or incorrect steps that are able to still steer the model towards the correct response on some training problems, but derail it otherwise. In this section, we present a conceptual model for constructing negatives that enables us to perform *per-step credit assignment*, and show that this approach can help us address these failure modes of positive data. We show that per-step DPO from Section 3 is a variant of this more general approach. We will then analyze scaling laws with negative data and empirically demonstrate that carefully constructed negative data can address issues with memorization. Finally, we theoretically prove that negative data improves sample-efficiency of $\mathcal{D}_{\mathrm{syn}}$.

### 6.1 Conceptual Model: Constructing Negatives to Enable Per-Step Credit Assignment

While naïvely contrasing an entire positive response $+y$ against an entire negative response $-y$ will increase the likelihood of *each* step that appears in $+y$ (even when incorrect or irrelevant) and reduce likelihood on each step appearing in $-y$ (even when accurate and relevant), it does not account for the importance of each step. Formally, given a negative solution trace $-y$, we would want to identify the first *critical* step where the model introduces a flaw $-y$, and emphasize alternate correct completions from this step that the model could have still produced. Likewise, given a positive solution trace, $+y$, we would like to identify if a given step $+y_i$ does not make progress towards the solution by identifying if there exist alternatives from its predecessor step, $+y_{1:i-1}$, which now presents a key decision-making point. **What are these critical steps and how can we identify them procedurally?**

**Value functions.** We can formalize this notion of a critical step under the notion of value functions from reinforcement learning (RL). Recall that both $+y$ and $-y$ are sampled from $\pi_{\mathrm{sft}}$. For problem $x$, with correct solution $y$, a response $\hat{y}$ with a sequence of steps $\hat{y}_{1:i-1}$, and a candidate step $\hat{y}_i$, we define the value function for step $y_i$, and previous steps under some policy $\tilde{\pi}$ as:

$$Q_{\tilde{\pi}}(\underbrace{x, \hat{y}_{1:i-1}}_{\text{state}}, \underbrace{\hat{y}_i}_{\text{action}}) = \underbrace{\mathbb{E}_{y_{i+1:L}^{\mathrm{new}} \sim \tilde{\pi}(\cdot|x,\hat{y}_{1:i})} \Big[ r \big( [\hat{y}_{1:i}, y_{i+1:L}^{\mathrm{new}}], y \big) \Big]}_{\text{expected future reward under new actions sampled by policy } \tilde{\pi}} \tag{2}$$

Intuitively, for any partial solution upto $i$ steps, this Q-function evaluates the probability of succeeding at solving the problem given the remaining budget of $L - i$ more steps, in expectation over all possible futures sampled from some policy $\tilde{\pi}$. Our conceptual model treats the policy $\tilde{\pi}$ as an algorithmic design choice that can differ for algorithms using negative data. As we see later, choosing $\tilde{\pi}$ as the Best-of-K distribution around $\pi_{\mathrm{sft}}$ (denoted as $\mathrm{BoK}(\pi_{\mathrm{sft}})$) enables a particularly interesting tradeoff between $Q$-value estimation and policy improvement. Another common choice is $\pi_{\mathrm{sft}}$ itself. Now,

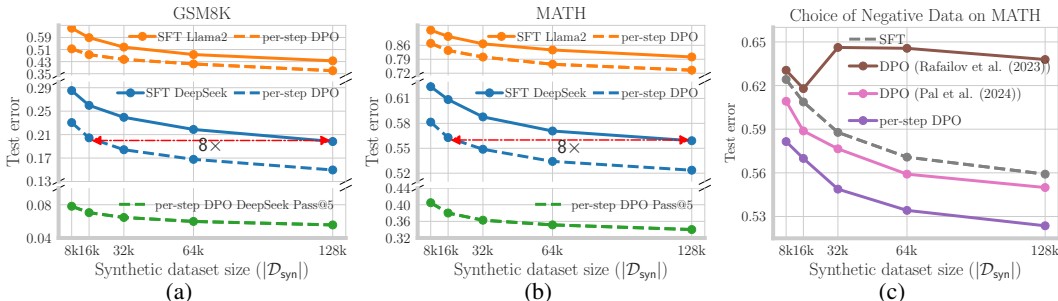

**Figure 5:** *Negative data scaling laws:* We evaluate algorithms that consume negative data as we scale $\mathcal{D}_{\mathrm{syn}}$, and compare them with only positive training (SFT) on $\mathcal{D}_{\mathrm{syn}}$. On GSM8K (a) and MATH (b), we observe an $8\times$ gain from per-step DPO (Section 4) which aligns with our model of negative data that enables per-step credit assignment. In (c) we compare different negative data construction algorithms, and particularly note that naïvely pairing positives and negatives [41] leads to worse performance as we scale $\mathcal{D}_{\mathrm{syn}}$.

for any given step $\hat{\boldsymbol{y}}_i$, we can define its *advantage* as the relative change in $Q_{\tilde{\pi}}$ when adding step $\hat{\boldsymbol{y}}_i$ in comparison with other possible candidates for step $i$ as follows:

$$A_{\tilde{\pi}}(\boldsymbol{x}, \hat{\boldsymbol{y}}_{1:i-1}; \hat{\boldsymbol{y}}_i) = Q_{\tilde{\pi}}(\boldsymbol{x}, \hat{\boldsymbol{y}}_{1:i-1}, \hat{\boldsymbol{y}}_i) - Q_{\tilde{\pi}}(\boldsymbol{x}, \hat{\boldsymbol{y}}_{1:i-2}, \hat{\boldsymbol{y}}_{i-1}). \quad (3)$$

Equation 3 is identical to the definition of advantage of an action (*i.e.*, $\hat{\boldsymbol{y}}_i$) at a state ($\boldsymbol{x}, \hat{\boldsymbol{y}}_{1:i-1}$) from RL [53], in that it is the gap between the Q-value of a state-action pair and the value function of the state (which itself is equal to the Q-value of the *previous* step due to deterministic dynamics).

**Critical steps, per-step DPO, and advantage-weighted RL.** We can use advantages (Equation 3) to characterize critical steps. Steps that attain a higher advantage value than others are **critical** since need to be generated more precisely to solve the problem. In contrast, steps that with very low advantage values are likely worse and must be unlearned. Our definition of the advantage function implies that one can calculate advantages for each step in a response via additional Monte Carlo rollouts starting from prefixes defined by partial solutions. One could then use these advantage estimates

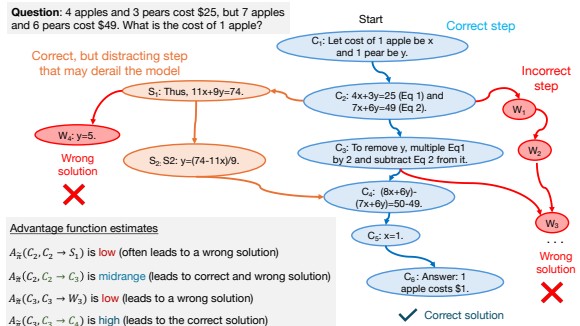

**Figure 6:** Illustration of advantage estimation from negative data on a didactic example in synthetic model generations. Critical steps are those with high advantage values.

(Equation 3) for training the model, for example, by running advantage-weighted reinforcement learning [40]. An alternate option would be to skip the computation of advantage estimates but instead rely on implicit approaches that optimize the advantage-weighted objective without computing their values. Theorem 6.1 shows that DPO performed over a precise pair distribution contrasting positive and negative traces obtained via additional rollouts from $\tilde{\pi}$, on prefixes of a response sampled from $\pi_{\mathrm{sft}}$ is equivalent to advantage-weighted RL. A proof of Theorem 6.1 is in Appendix E. Note that unlike the standard reduction of DPO to the RL objective under *some* reward function [41, 42], Theorem 6.1 is stronger in that it identifies the value function induced by per-step DPO.

**Theorem 6.1** (Equivalence of advantage-weighted RL and DPO with per-step pairs). *The optimal policy from Equation 1 with $\mathcal{D}^{\pm}_{\pi_{\mathrm{sft}}}$ given by $(\boldsymbol{x}, [\boldsymbol{y}_{1:i}, +\boldsymbol{y}_{i+1}], [\boldsymbol{y}_{1:i}, -\boldsymbol{y}_{i+1}])$ where the positive and negative traces share prefix $\boldsymbol{y}_{1:i} \sim \pi_{\mathrm{sft}}$, and $-\boldsymbol{y}_{i+1} \sim \pi_{\mathrm{sft}}(\cdot|\boldsymbol{x}, \boldsymbol{y}_{1:i})$, $+\boldsymbol{y}_{i+1} \sim \sigma(A_{\tilde{\pi}}(\boldsymbol{x}, \boldsymbol{y}_{1:i}; \cdot) - A_{\tilde{\pi}}(\boldsymbol{x}, \boldsymbol{y}_{1:i}; -\boldsymbol{y}_{i+1}))$, is identical to the optima of the advantage-weighted RL objective:*

$$\max_{\pi} \ \mathbb{E}_{\boldsymbol{x} \sim p_{\mathrm{syn}}(\boldsymbol{x}), \boldsymbol{y} \sim \pi_{\mathrm{sft}}(\cdot|\boldsymbol{x})} \left[ \sum_{i=1}^{L} \log \pi(\boldsymbol{y}_i | \boldsymbol{x}, \boldsymbol{y}_{0:i-1}) \cdot \exp\left(A_{\tilde{\pi}}(\boldsymbol{x}, \boldsymbol{y}_{0:i-1}; \boldsymbol{y}_i)/\beta\right) \right]. \quad (4)$$

**Practical instantation of DPO with per-step pairs.** In most of our experiments, we instantiate a practical version of the above framework, following the scheme in Hwang et al. [23]. This is a special case (Part 1) of the complete algorithm shown in Algorithm 1 (see Appendix B). Unless otherwise mentioned, we use "per-step DPO" to refer to this version (Part 1 only) in practice. We will also experiment with the complete version (parts 1 and 2) later in Section 6.3.3. Instead of computing

advantage estimates for each step, and then sampling preference pairs, as described in Theorem 6.1, we approximate this by only Q-value estimates on 8 negative responses for each question in the synthetic dataset, with $\tilde{\pi}$ chosen to be the best-of-K policy, $\text{BoK}(\pi_{\text{sft}})$ where $K = 5$. There are two benefits associated with this choice of $\tilde{\pi}$, especially a higher value of $K$: **(i)** estimating the advantage in Equation 3 with Monte Carlo rollouts exhibits lower variance when $K$ is large, since a larger budget $K$ would lead most steps to have higher Q-values and the variance of Bernoulli reduces as Q-value $\to 1$; and **(ii)** $Q_{\text{BoK}(\pi_{\text{sft}})}$ is a non-decreasing function in $K$ for any state-action, which implies that the solution of advantage-weighted RL objective, in principle, can now improve over a better policy $\text{BoK}(\pi_{\text{sft}})$, compared to $\pi_{\text{sft}}$. Next, we discuss scaling results for negative data, and then in Section 6.3 show how per-step credit assignment improves generalization and suppresses irrelevant and incorrect steps appearing in a response, extracting more gains from the same synthetic data.

## 6.2 Scaling Results for Negative Data

Observe in Figure 5(a,b), that for both DeepSeek-Math-7B and LLama2-7B models, per-step DPO improves performance beyond the SFT policy and the performance continues to scale favorably as data size increases. In fact, for any given size of $\mathcal{D}_{\text{syn}}$, per-step DPO also substantially improves over RFT (Figure 2) on both datasets, and overall, **while RFT improved effective data size of $\mathcal{D}_{\text{syn}}$ by $2\times$, additionally training on negative data extends the performance improvement to $8\times$ the size of $\mathcal{D}_{\text{syn}}$**. Additionally, since per-step DPO estimates advantage of each step under the Best-of-5 policy, one might expect a saturation in the pass@5 performance of the per-step DPO solution. On the contrary, we find that pass@5 performance also improves consistently. In Appendix D we present results for a filtered version of RFT. Here, steps with high advantages from positive/negative data are cloned. This resolves the scaling issue seen when naively scaling positive data in Figure 2(c).

**Choice of negative data matters.** In Figure 5(c) we plot negative data scaling laws where the choice of negative data (thereby pairs for DPO in Equation 1) differs. Observe that standard pairing of positive and negative responses in $\mathcal{D}_{\pi_{\text{sft}}}^{\pm}$ for DPO [41] does not improve over the SFT policy. As such, we tuned $\beta$ in Equation 1 for DPO but could not fully avoid performance degradation. Our conceptual model explains this result: contrasting arbitrary positives and negatives would result in an incorrect induced advantage function, training with DPO will exacerbate spurious correlations that maximize this induced advantage function [39, 46, 64]. In fact, Pal et al. [38] also find similar concerns with random pairing and instead pair positives and negatives with highest edit distance, which leads to some improvement, but still performs poorer than per-step DPO that accounts for credit.

> **Takeaways for scaling negative synthetic data**
>
> - Negative data can identify high-advantage (critical) steps in model-generated responses.
> - We can construct negative data distribution that equates DPO to advantage-weighted RL. Negative data used in this way improves the sample efficiency of synthetic data by $8\times$.

## 6.3 Why Does Credit Assignment from Negative Data Improve Model Generalization?

Our conceptual model illustrates that per-step DPO can perform credit assignment, and identify critical steps over irrelevant ones via advantage estimates. We saw that this improves test performance and scaling. Now, we attempt to understand why per-step credit assignment should improve generalization by understanding the generalization properties of advantage-weighted RL. We present two empirical studies below, and a formal theoretical guarantee combining these insights is shown in Appendix F.

### 6.3.1 Advantage-Weighted RL De-Emphasizes Spurious Steps and Emphasizes Critical Steps

Our key insight is that spurious correlations emerge in monolithic SFT or RFT due to the well-known issue of causal confusion [12] in imitation learning: by memorizing incorrect or irrelevant steps and associating them with the correctness of the final answer, the model fails to generalize on novel problems, as we saw in Figure 4. We now explain how *online* model-specific interventions and advantage estimation would address this issue. Consider $\tilde{\pi} = \pi_{\text{sft}}$. As we show later, in under-trained models memorized steps are imperfectly cloned under $\pi_{\text{sft}}$, implying that while teacher-forcing loss is low for some spurious, memorized step $\boldsymbol{y}_s$, sampling paths from $\pi_{\text{sft}}$, conditioned on $\boldsymbol{y}_{1:s}$ is likely to generate incorrect responses. This means $\boldsymbol{y}_s$ attains a low advantage. On the other

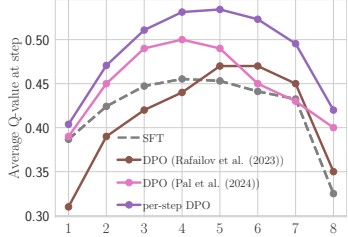

**Figure 7:** Per-step DPO improves Q-values at each step, standard DPO only improves at irrelevant steps.

hand, for a correct step, *whp* estimated advantage is higher. Thus, training the model with advantage weighted RL would de-emphasize spurious steps and emphasize critical steps. Running per-step DPO on data generated by the RFT model that has overfit on spurious correlations improves accuracy by >6% (Figure 4). We visualize advantages in Appendix K. In Figure 7, we plot the average Q-value of a step for different negative data schemes, and note that only per-step DPO improves over SFT at each step, as expected based on the connection to advantage-weighted RL (Theorem 6.1). Standard DPO fails to improve performance since it has poor success rate at earlier (critical) steps.

### 6.3.2 Why Does Generalization Improve?: Connecting Advantage-Weighted RL to DRO

In the previous section, we discussed how advantage-weighted RL preferentially weighs the next-token prediction loss at each step. Now, we attempt to conceptually understand why this could improve generalization. For this, we present an intuitive explanation by drawing a connection between advantage-weighted RL and a distributionally robust optimization (DRO) algorithm, named Group DRO, commonly used to improve worst-group robustness in supervised learning [43].

**Intuitive explanation.** During inference, the SFT policy can fail even on training problems, especially in scenarios where the SFT policy has failed to perfectly clone the next step at each intermediate step in the SFT data. As previously discussed, these steps also present with low advantage values. One way to reduce the chance of compounding inference time errors [45] is to preferentially minimize the negative log-likelihood loss *more* for the critical step, i.e., those steps from where the model is more likely to arrive at a wrong answer. If we iteratively update the policy with gradient steps computed over a re-weighted next-step prediction objective where each step is weighted by its advantage estimate, then the resulting algorithm intuitively exhibits this characteristic similarly to distributionally robust optimizers (DRO) [28]. Similar to how DRO solutions guarantee that all subpopulations – both majority and minority subpopulations – in the training data achieve low loss values, the solution for the advantage-weighted RL objective guarantees that the negative log-likelihood of the critical steps with high advantage estimates under $\tilde{\pi}$ (which of per-step DPO is $\text{BoK}(\pi_{\text{sft}})$) is also low, to a similar extent as the other more prevalent non-critical steps.

In other words, **our insight** is that weighting steps using advantages in Equation 4 upweights the likelihood of the underrepresented critical states while down-weighting it for the spurious ones. The guarantees on the training data ($\mathcal{D}_{\text{syn}}$) also translate to the population level objective when the weights for on-policy samples (advantage estimates) are accurate [49] and the policy is sufficiently regularized [47]. Since correct behavior at critical steps determine the correctness of the overall solution, an elevated degree of correctness at executing critical steps at the population level implies a higher test accuracy on the reasoning task.

### 6.3.3 But, Attaining Low Generalization Error Requires Low Advantage Estimation Error

The practical efficacy of algorithms that use negative data for credit assignment requires the advantage estimation error to be low with fewer rollouts from $\tilde{\pi}$. For discussion, consider $\tilde{\pi} = \pi_{\text{sft}}$. When the initial advantage of a spurious step is incorrectly over-estimated, negative data algorithms up-weight the likelihood further. This only leads to further memorization. Hence, most Monte-Carlo rollouts from $\pi_{\text{sft}}$ would rely upon the memorized feature. Since the model generates the correct answer from the memorized feature, it would estimate higher $A_{\pi_{\text{sft}}}$, and this downward spiral of training with increasing weights on the spurious step leads to test-time model collapse. On the other hand, when $\tilde{\pi} = \text{BoK}(\pi_{\text{sft}})$ for a higher value of $K$, the Monte-Carlo advantage estimator has a lower variance (and error). This discussion also justifies the choice of $K{=}5$, an intermediate value, in per-step DPO.

### 6.3.4 Validating Claims About Generalization: Controlled Analysis on a Didactic Problem

With the above insights, we now study the influence of $\pi_{\text{sft}}$ on the generalization effects of per-step DPO. For our analysis, we consider a didactic star graph problem (Appendix I) from Bachmann and Nagarajan [4], where given a graph in the shape of a star and a query (center/end node), the model is asked to output the full path between the start/end nodes. This task highlights the failure of SFT at planning problems (akin to math reasoning). They show that $\pi_{\text{sft}}$ minimizes SFT loss by memorizing the "hard-to-predict" node adjacent to the center, and copying the rest from the input graph. It is clear that the failure stems from not being able to identify the critical adjacent token. We will show how credit assignment with negative data accurately upweights the critical token and unlearns the memorized token. To vary the choice of $\pi_{\text{sft}}$, we choose several intermediate checkpoints obtained during supervised finetuning for synthetic negative data generation. We consider three initializations: **(1)** an under-trained SFT model with a large training and test loss, and **(2)** an SFT model obtained by

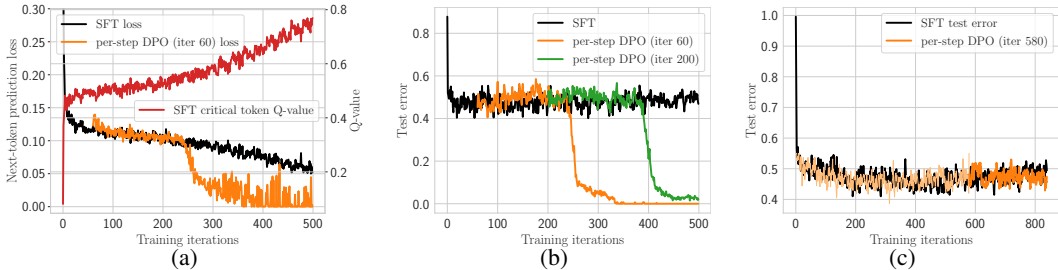

**Figure 8:** *Didactic analysis on star graph:* In (a) we plot the SFT loss and Q-value of the critical token (adjacent node) for SFT and per-step DPO (starting from iter 60). Indicative of memorization SFT loss decreases at a slow rate, matching the slow rate of increase in the Q-value. In contrast per-step DPO loss sharply decreases during training. In (b) we notice a corresponding phase transition in the test error of per-step DPO starting from different under-trained SFT checkpoints, which does not happen for an over-trained SFT checkpoint in (c).

early-stopping based on a held-out validation set, where the validation loss is the lowest, and **(3)** an over-trained SFT checkpoint, with a low training but high validation loss.

**(1) & (2): Training on negative data from an under-trained or early-stopped $\pi_{\text{sft}}$ improves both training loss and test performance.** As shown in Figure 8(a,b), we find that when training with negative data from iteration 60 (under-trained $\pi_{\text{sft}}$) and iteration 200 (early-stopped $\pi_{\text{sft}}$), utilizing per-step DPO reduces the training loss very aggresively. These benefits translate to test losses and performance as well (Figure 8(b), orange and green). In contrast, supervised finetuning exhibits a nearly-flat test loss landscape, although the train loss reduces slowly. Upon a closer inspection, we find that training on positive data via SFT only tends to memorize the critical token in the training data using non-generalizable features, and hence, the resulting model does not generalize to novel problems. More training with SFT is unable to "unlearn" this spurious correlation and does not reduce the loss function. On the other hand, per-step DPO with negative data is able to unlearn this spurious feature and drives improvement, as evident by the drastic improvement on train and test.

**(3) Training on negative data from an over-trained SFT initialization leads to model collapse.** When training with negative data on an over-trained $\pi_{\text{sft}}$ (iteration 580) in Figure 8(c), we observe that both SFT and per-step DPO exhibit identical test errors since training with more negative data simply exacerbates the model's dependence on memorizing the critical token, which manifests in the form of lower test losses. This is also an example where Monte-Carlo samples from the over-trained checkpoint estimates a high advantage since Q-value is already high at iteration 500 (in (a)). This means that when the SFT policy has sufficiently memorized the training data using a spurious feature, training further is unable to unlearn this dependence. Hence, we find that in this regime, negative data leads to no improvement, capping performance at what was attained by fine-tuning on positive data.

> **Takeaways for generalization and spurious correlations with negative data**
>
> Advantage-weighted RL unlearns spurious steps and improves generalization when: (i) advantage estimation error is low; and (ii) the model is under-trained enough that imperfectly cloned spurious steps have low advantage, which can then be estimated with negative data.

## 7 Discussion and Conclusion

Our work studies the role of synthetic data for improving math reasoning capabilities of LLMs. We find that while the typical approach of collecting new questions and corresponding positive (correct) solutions from capable models like GPT-4/Gemini-1.5 presents underwhelming data scaling. The sample efficiency of the same data can be improved up to 2× by sampling more positive traces from the 7B sized models SFT-ed on the original data. However, training on positive self-generated synthetic data alone often amplifies the model's dependence on spurious steps, that erroneously appear to lead to a good solution but do not generalize to novel problems and hurt test performance. That said, surprisingly, we show that negative (incorrect) traces sampled from the same SFT model can be used to address the failure modes of training on only positive data. In particular, negative data can be used to estimate advantage values for every step, and using these advantage estimates via RL enables us to address this problem. We show how the advantages can be used implicitly by preference optimization objectives. We show how training on an instance of this objective leads to 8× improvements in sample efficiency of the synthetic data used.

## Acknowledgements

This work was done at CMU. We thank Vaishnavh Nagarajan, Yi Su, Aleksandra Faust, Hyeonbin Hwang, Christina Baek, Charlie Snell, Seohong Park, Gaurav Ghosal, Aditi Raghunathan, Katie Kang, Don Dennis, Dhruv Malik, and Pratiksha Thaker for informative discussions and feedback on an earlier version of this paper. This work was supported by compute donations from Google Cloud (TRC) and MultiOn. AS thanks OpenAI and Google respectively for providing GPT4-Turbo and Gemini-1.5 Pro credits for academic use. AK and YG thank Tianhe Yu for feedback on the paper. This work was supported in part by the National Science Foundation grants IIS2145670 and CCF2107024, and funding from Amazon, Apple, Google, Intel, Meta, and the CyLab Security and Privacy Institute. Any opinions, findings and conclusions expressed in this material are those of the author(s) and do not necessarily reflect the views of any of these funding agencies.

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

# Appendices

## A   Limitations of our Work

While our work provides some results and conceptual models to understand the role of synthetic data for reasoning, there are still many open questions that need to be answered to fully understand its utility. While synthetic data from LLMs like Gemini and GPT-4 holds great potential, for more complex reasoning problems (more complicated than the datasets evaluated in our work), synthetic data generated from more capable models can contain errors. Generating negative/positive data by referencing synthetic data answers can reinforce unwanted spurious correlations highlighted in our work. This means that novel recipes for generating synthetic problems may be utilized in the future, and our analysis might need to be re-done. That said, we believe that our insights about algorithmic behavior with synthetic data are still quite general and should transfer to these novel settings as well. Ultimately, we would want that training on synthetic data improves transfer and generalization abilities of the model in general reasoning scenarios, and to this end, an evaluation of transfer capabilities is an important avenue that future work should focus on.

## B   Per-step DPO Algorithm

---

**Algorithm 1** Per-step DPO (Part 1: Practical version for most experiments; Parts 1 + 2: Complete version)

---

**Require:** Synthetic dataset: $\mathcal{D}_{\mathrm{syn}}$, SFT policy finetuned on $\mathcal{D}_{\mathrm{syn}}$: $\pi_{\mathrm{sft}}$, sampling policy $\tilde{\pi}$.

1:  Initialize per-step DPO dataset $\mathcal{D}^{\pm}_{\pi_{\mathrm{sft}}} \leftarrow \{\}$.
2:  **for** $(\boldsymbol{x}, \boldsymbol{y}) \in \mathcal{D}_{\mathrm{syn}} \cup \mathcal{D}^{+}_{\pi_{\mathrm{sft}}}$ **do**
3:      # **Part 1:** Identify critical steps in incorrect responses
4:      Sample multiple incorrect answers $-\hat{\boldsymbol{y}} \sim \pi_{\mathrm{sft}}(\cdot \mid \boldsymbol{x})$, and collect them in set $\mathcal{C}(\boldsymbol{x})$.
5:      **for** $-\hat{\boldsymbol{y}} := [-\hat{\boldsymbol{y}}_1, \ldots, -\hat{\boldsymbol{y}}_L] \in \mathcal{C}(\boldsymbol{x})$ **do**
6:          Compute the Monte Carlo estimate for $Q_{\tilde{\pi}}(\boldsymbol{x}, -\hat{\boldsymbol{y}}_{1:i-1}; -\hat{\boldsymbol{y}}_i)$ for each step $-\hat{\boldsymbol{y}}_i$.
7:          If $-\hat{\boldsymbol{y}}_c$ is the first step with least $Q_{\tilde{\pi}}(\boldsymbol{x}, -\hat{\boldsymbol{y}}_{1:i-1}; -\hat{\boldsymbol{y}}_i)$, then $\mathcal{D}^{\pm}_{\pi_{\mathrm{sft}}} \leftarrow \mathcal{D}^{\pm}_{\pi_{\mathrm{sft}}} \cup \{(\boldsymbol{x}, \boldsymbol{y}, -\hat{\boldsymbol{y}}_{1:c})\}$.
8:      **end for**
9:      # **Part 2:** Identify spurious steps in correct responses
10:     Sample multiple correct answers $+\hat{\boldsymbol{y}} \sim \pi_{\mathrm{sft}}(\cdot \mid \boldsymbol{x})$, and collect them in set $\mathcal{C}'(\boldsymbol{x})$.
11:     **for** $+\hat{\boldsymbol{y}} := [+\hat{\boldsymbol{y}}_1, \ldots, +\hat{\boldsymbol{y}}_L] \in \mathcal{C}'(\boldsymbol{x})$ **do**
12:         Compute the Monte Carlo estimate for $Q_{\tilde{\pi}}(\boldsymbol{x}, +\hat{\boldsymbol{y}}_{1:i-1}; +\hat{\boldsymbol{y}}_i)$ for each step $+\hat{\boldsymbol{y}}_i$.
13:         If $+\hat{\boldsymbol{y}}_c$ is the first step with least $Q_{\tilde{\pi}}(\boldsymbol{x}, +\hat{\boldsymbol{y}}_{1:i-1}; +\hat{\boldsymbol{y}}_i)$, then $\mathcal{D}^{\pm}_{\pi_{\mathrm{sft}}} \leftarrow \mathcal{D}^{\pm}_{\pi_{\mathrm{sft}}} \cup \{(\boldsymbol{x}, \boldsymbol{y}, +\hat{\boldsymbol{y}}_{1:c})\}$.
14:     **end for**
15: **end for**
16: Optimize DPO loss in Equation (1) on $\mathcal{D}^{\pm}_{\pi_{\mathrm{sft}}}$ with $\pi_{\mathrm{sft}}$ as the reference policy.

---

## C   Additional Experiments using Positive Synthetic Data (Section 5)

Recall from Section 5 we observed a 2× boost in sample efficiency (over $\pi_{\mathrm{sft}}$) of the question/answer pairs in the synthetic data when we cloned positive solutions sampled from $\pi_{\mathrm{sft}}$. Note that one may also attribute better performance of RFT to improved coverage from multiple responses in $\mathcal{D}^{+}_{\pi_{\mathrm{sft}}}$ for each question in $\mathcal{D}_{\mathrm{syn}}$. We find that even when RFT data is restricted to one solution per question, base LLMs finetuned on it outperform SFT consistently by $> 1\%$. In Figure 9, we plot the performance of DeepSeek-Math-7B finetuned on SFT data $\mathcal{D}_{\mathrm{syn}}$ and RFT data $\mathcal{D}^{+}_{\pi_{\mathrm{sft}}}$ where $\mathcal{D}^{+}_{\pi_{\mathrm{sft}}}$ has the same questions as $\mathcal{D}_{\mathrm{syn}}$, and only one positive solution per question, sampled from $\pi_{\mathrm{sft}}$ finetuned on $\mathcal{D}_{\mathrm{syn}}$. Thus, both SFT and RFT datasets are of the same size. This means that a significant portion of the 2× sample efficiency gains we observe for RFT in Figure 2(a,b) can be attributed to RFT data

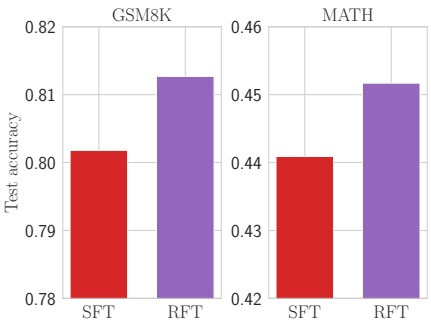

**Figure 9:** RFT data with a single (self-generated) correct solution per problem outperforms SFT data (from highly-capable models) of the same size.

from $\pi_{\mathrm{sft}}$ being easier-to-fit, and not purely because RFT data improves coverage by finetuning on multiple solution traces per question.

# D  Additional Experiments using Negative Synthetic Data (Section 6)

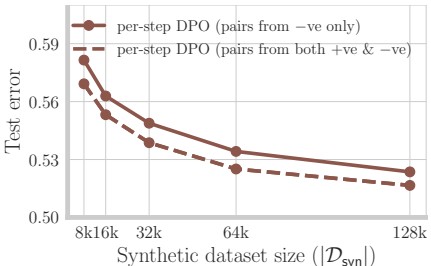

**Figure 10:** On MATH, improving advantage estimates by computing advantages over both positive and negative traces sampled from $\pi_{\mathrm{sft}}$ improves estimation error and final performance for per-step DPO.

**Improving per-step DPO [23] with a closer approximation of advantage-weighted RL (Parts 1 + 2 in Algorithm 1).** Now, we discuss an experiment that improves the performance of per-step DPO [23] by running the full version of Algorithm 1. In particular, we add new preference pairs to the dataset of per-step DPO algorithm starting from positive samples. Recall from Section 4 and Algorithm 1, that for a problem $x$, with correct answer $y$ given by SFT or RFT data, Part 1 of per-step DPO estimates the expected accuracy (Q-value) of each step in a negative rollout $-\hat{y} \sim \pi_{\mathrm{sft}}(\cdot \mid x)$ sampled from the SFT policy. For each step $-\hat{y}_c$ the Q-value is computed conditioned on $x$ and previous steps $-\hat{y}_{1:c-1}$. It then adds the triplet $(x, y, -\hat{y}_{1:c})$ to the DPO dataset. We improve the coverage and accuracy of advantage estimates via Part 2, i.e., repeating this process for steps appearing on a positive trace $+\hat{y} \sim \pi_{\mathrm{sft}}(\cdot \mid x)$ as well. Specifically, we add $(x, y, +\hat{y}_{1:c})$ to the DPO dataset, where the step $+\hat{y}_c$ is the first step in the positive trace to have a low Q-value (as dicated by a relative threshold on the Q-value), which indicates that $+\hat{y}_c$ is likely a spurious step that the SFT model generated. For individual steps that are more likely to occur in either positive or negative traces we improve coverage of alternate possible steps, and for steps that occur in both positive and negative traces, we lower the variance (and error) of the advantage estimate. In Figure 10, we compare the performance of per-step DPO runs with the datasets constructed from only negative vs. both positive and negative traces, and find that the latter has a lower test error for all sizes of $\mathcal{D}_{\mathrm{syn}}$.

**Advantage filtered per-step RFT.** We ran an experiment with advantage filtering on all the steps present in both positive and negative data from the SFT policy and cloned the filtered data. For this, we cloned responses with high advantage steps from positive and negative responses sampled from the SFT policy. We filter all steps where the minimum advantage across all steps is in the bottom 50% percentile. This "per-step RFT" outperforms standard RFT (Figure 11), indicating that training on useful steps from negative data can improve beyond only training on positive data alone. While per-step RFT is worse than per-step DPO, we believe that this only further hints at the point that even using low advantage steps (that per-step RFT filters) for training, can further improve.

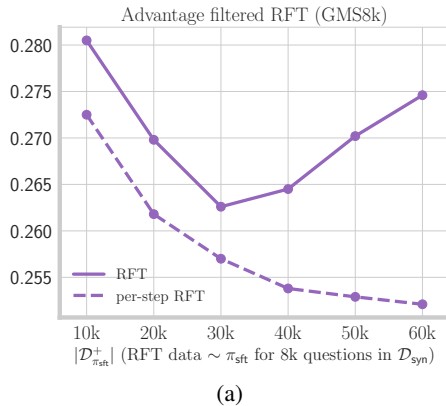
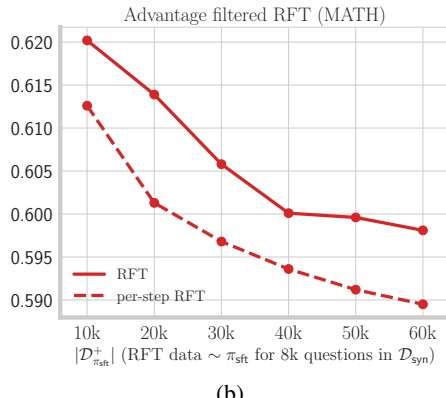

(a)                  (b)

**Figure 11:** *Advantage filtered RFT:* We clone responses with high advantage steps from positive and negative responses sampled from the SFT policy. We filter all responses where the minimum advantage across all steps is in the bottom 50% percentile.

## E    Proof of Theorem 6.1

We first restate the theorem statement and then provide a proof for this below. Our main goal in this theorem is to show that training with per-step DPO is equivalent to running advantage-weighted RL shown in the theoretical result.

**Theorem E.1** (Equivalence of advantage-weighted RL and DPO with per-step pairs)**.** *The optimal policy from Equation 1 with $\mathcal{D}_{\pi_{\text{sft}}}^{\pm}$ given by $(\boldsymbol{x}, [\boldsymbol{y}_{1:i}, +\boldsymbol{y}_{i+1}], [\boldsymbol{y}_{1:i}, -\boldsymbol{y}_{i+1}])$ where the positive and negative traces share prefix $\boldsymbol{y}_{1:i} \sim \pi_{\text{sft}}$, and $-\boldsymbol{y}_{i+1} \sim \pi_{\text{sft}}(\cdot|\boldsymbol{x}, \boldsymbol{y}_{1:i})$, $+\boldsymbol{y}_{i+1} \sim \sigma(A_{\tilde{\pi}}(\boldsymbol{x}, \boldsymbol{y}_{1:i}; \cdot) - A_{\tilde{\pi}}(\boldsymbol{x}, \boldsymbol{y}_{1:i}; -\boldsymbol{y}_{i+1}))$, is identical to the optima of the advantage-weighted RL objective:*

$$\max_{\pi} \; \mathbb{E}_{\boldsymbol{x} \sim p_{\text{syn}}(\boldsymbol{x}), \boldsymbol{y} \sim \pi_{\text{sft}}(\cdot|\boldsymbol{x})} \left[ \sum_{i=1}^{L} \log \pi(\boldsymbol{y}_i | \boldsymbol{x}, \boldsymbol{y}_{0:i-1}) \cdot \exp\left( A_{\tilde{\pi}}(\boldsymbol{x}, \boldsymbol{y}_{0:i-1}, \boldsymbol{y}_i)/\beta \right) \right]. \tag{5}$$

*Proof.* To prove this statement, we make the following observation: DPO [41] is equivalent to optimizing a KL-divergence penalized expected reward objective in an induced Bradly-Terry model of preferences defined by the reward function. That is, for any reward function $r(\boldsymbol{x}, \boldsymbol{y})$ over contexts $\boldsymbol{x} \sim \mu$ and responses $\boldsymbol{y}$, the optimal solution to the following RL objective:

$$\max_{\pi} \; \mathbb{E}_{\boldsymbol{x} \sim \mu, \boldsymbol{y} \sim \pi(\cdot|\boldsymbol{x})} \left[ r(\boldsymbol{x}, \boldsymbol{y}) \right] - \beta D_{\text{KL}}(\pi(\cdot|\boldsymbol{x}) || \pi_{\text{sft}}(\cdot|\boldsymbol{x})), \tag{6}$$

is given by the following advantage-weighted optimal policy, $\pi^*(\cdot|\cdot)$:

$$\forall \boldsymbol{x}, \boldsymbol{y}, \; \pi^*(\boldsymbol{y}|\boldsymbol{x}) \propto \pi_{\text{sft}}(\boldsymbol{y}|\boldsymbol{x}) \cdot \exp\left( \frac{r(\boldsymbol{x}, \boldsymbol{y})}{\beta} \right), \tag{7}$$

and one can learn this optimal policy by running DPO on preference tuples $(\boldsymbol{x}, \boldsymbol{y}_1, \boldsymbol{y}_2)$ sampled by the Bradly-Terry model [6] induced by the reward function $r$:

$$p(\boldsymbol{y}_1 \succcurlyeq \boldsymbol{y}_2 | \boldsymbol{x}) = \frac{\exp(r(\boldsymbol{x}, \boldsymbol{y}_1))}{\exp(r(\boldsymbol{x}, \boldsymbol{y}_1)) + \exp(r(\boldsymbol{x}, \boldsymbol{y}_2))}. \tag{8}$$

Given this background information, we know that the optimal advantage-weighted RL policy optimizing Equation 5 is given by:

$$\forall \boldsymbol{x}, \boldsymbol{y}_{0:i}, \; \pi(\boldsymbol{y}_i | \boldsymbol{x}, \boldsymbol{y}_{0:i-1}) \propto \pi_{\text{sft}}(\boldsymbol{y}_i | \boldsymbol{x}, \boldsymbol{y}_{0:i-1}) \cdot \exp\left( \frac{A_{\tilde{\pi}}(\boldsymbol{x}, \boldsymbol{y}_{0:i-1}, \boldsymbol{y}_i)}{\beta} \right). \tag{9}$$

Combining Equation 9 with the equivalence between Equation 7 and the Bradly-Terry model (Equation 8), we obtain that, if preference pairs $(\boldsymbol{x}, [\boldsymbol{y}_{1:i}, +\boldsymbol{y}_{i+1}], [\boldsymbol{y}_{1:i}, -\boldsymbol{y}_{i+1}])$ were sampled from the SFT policy: $+\boldsymbol{y}_{i+1} \sim \pi_{\text{sft}}(\cdot|\boldsymbol{x}, \boldsymbol{y}_{0:i})$ and $-\boldsymbol{y}_{i+1} \sim \pi_{\text{sft}}(\cdot|\boldsymbol{x}, \boldsymbol{y}_{0:i})$, and labeled according to Equation 8 applied on advantage estimates, then we obtain the desired equivalence result. $\square$

# F   Theory: Why Does Negative Data Improve Generalization?

We saw in Section 6.3 that collecting negative data from an appropriate SFT policy $\pi_{\text{sft}}$ and an appropriate $K$, and training on this data improves generalization performance of the resulting model. In this section, building on the equivalence to advantage-weighted RL (Theorem 6.1), we attempt to formalize this observation into a performance guarantee. In particular, we show below that training on negative data implies that we are able to improve over the SFT policy, especially via the detection of critical steps, that attain high advantages, $A_{\tilde{\pi}}(\boldsymbol{x}, \boldsymbol{y}_{0:i-1}, \boldsymbol{y}_i)$, that are otherwise not prioritized by training on positive data alone. Our theoretical result extends guarantees from the RL literature [27] comparing RL with imitation learning to show that indeed the use of RL (and hence negative data) improves over imitation alone.

**Notation and setup.**  Define the policy obtained after advantage-weighted RL training as $\pi_{\text{neg}}$. Concretely, $\pi_{\text{neg}}(\boldsymbol{y}|\boldsymbol{x})$ is given as:

$$\forall \boldsymbol{x}, \boldsymbol{y}_{0:j+1}, \pi_{\text{neg}}(\boldsymbol{y}_{j+1}|\boldsymbol{x}, \boldsymbol{y}_{0:j}) = \frac{1}{\widehat{\mathbb{Z}}(\boldsymbol{x}, \boldsymbol{y}_{0:j})} \pi_{\text{sft}}(\boldsymbol{y}_{j+1}|\boldsymbol{x}, \boldsymbol{y}_{0:j}) \cdot \exp\left( \frac{\hat{A}_{\tilde{\pi}}(\boldsymbol{x}, \boldsymbol{y}_{0:j}, \boldsymbol{y}_{j+1})}{\beta} \right), \quad (10)$$

where the normalization factor is given by $\mathbb{Z}(\boldsymbol{x}, \boldsymbol{y}_{0:j})$ for each of the per-step policy distributions. This normalization factor is a critical factor that will drive the core of the theoretical result. We also note that the normalization factor in Equation 10 is derived from *empirical* advantage estimates and not from the expected estimates for the advantage value. Following Agarwal et al. [2], Kumar et al. [27], we operate in a tabular setting with a discrete (but combinatorially-large and variable-length) action space of responses, and our proof follows Theorem 4.4 in Kumar et al. [27].

**Theorem F.1** (Utility of negative data over positive data.)**.**  *Let $\pi_{neg}$ denote the policy obtained after advantage-weighted RL (Equation 10) under an empirical distribution $\hat{\mu}$ over prompts $\boldsymbol{x}$. Then the difference between the expected reward (i.e., task success rate), $J(\cdot)$, attained by $\pi_{neg}$ and $\pi_{\text{sft}}$ is lower-bounded as:*

$$J(\pi_{neg}) - J(\pi_{\text{sft}}) \gtrsim \quad \beta \cdot \mathbb{E}_{\boldsymbol{x}_i \sim \hat{\mu}, \boldsymbol{y}_{i,0:L} \sim \pi_{neg}(\cdot|\boldsymbol{x}_i)} \left[ \sum_{j=1}^{L} \log \mathbb{Z}(\boldsymbol{x}_i, \boldsymbol{y}_{i,0:j}) \right]$$
$$- \textit{(overestimation in } \hat{A}_{\tilde{\pi}}(\boldsymbol{x}, \boldsymbol{y}_{0:i-1}, \boldsymbol{y}_i)) + \frac{c_0}{\sqrt{|\mathcal{D}_{\text{syn}}|}},$$

*where $\mathbb{Z}(\clubsuit, \circ)$ denotes the sum over exponentiated differences of the advantage and log likelihood values under $\pi_{\text{sft}}$ for all possible candidate steps given a problem $\clubsuit$ and a partial solution $\circ$. That is,*

$$\mathbb{Z}(\clubsuit, \circ) := \sum_{\spadesuit \in \textit{ step candidates}} \exp\left( \frac{A_{\tilde{\pi}}(\clubsuit, \circ; \spadesuit)}{\beta} + \log \pi_{\text{sft}}(\spadesuit|\clubsuit, \circ) \right),$$

*$c_0$ is a constant depending upon the Rademacher complexity of the space of policies $\pi_{neg}$ close to the SFT policy under the KL-divergence, $|\mathcal{D}_{\text{syn}}|$ denotes the size of synthetic training prompts.*

*Proof.*  To begin the proof, we recall that we are operating in a discrete action space of steps $\boldsymbol{y}_i$, although this space is exponentially large. Since we operate in discrete action spaces, we invoke Lemma 5 from Agarwal et al. [2] for analyzing softmax policy gradient methods (this Lemma was extended by Lemma B.11 from Kumar et al. [27] for comparing BC vs offline RL). Denote by $\hat{J}(\pi)$, the reward attained by policy $\pi$ in expectation over the empirical distribution $\hat{\mu}$:

$$\hat{J}(\pi_{\text{neg}}) - \hat{J}(\pi_{\text{sft}}) := \mathbb{E}_{\boldsymbol{x} \sim \hat{\mu}} \left[ \widehat{V}^{\pi_{\text{neg}}}(\boldsymbol{x}) \right] - \mathbb{E}_{\boldsymbol{x} \sim \hat{\mu}} \left[ \widehat{V}^{\pi_{\text{sft}}}(\boldsymbol{x}) \right] \geq \beta \mathbb{E}_{\boldsymbol{x} \sim \hat{\mu}} \left[ \log \widehat{\mathbb{Z}}(\boldsymbol{x}) \right]. \quad (11)$$

We utilize the performance difference lemma [25] on the MDP induced by the set of initial problems in the training distribution $\widehat{\mu}$, and the model induced deterministic dynamics distribution:

$$
\widehat{J}(\pi_{\text{neg}}) - \widehat{J}(\pi_{\text{sft}}) = \sum_{j=1}^{L} \mathbb{E}_{\boldsymbol{x} \sim \widehat{\mu}, \boldsymbol{y}_{0:j-1} \sim \pi_{\text{neg}}(\cdot|\boldsymbol{x})} \left[ \sum_{\boldsymbol{y}_j} \pi_{\text{neg}}(\boldsymbol{y}_j|\boldsymbol{x}, \boldsymbol{y}_{0:j-1}) \widehat{A}_{\tilde{\pi}}(\boldsymbol{x}, \boldsymbol{y}_{0:i-1}, \boldsymbol{y}_i) \right]
$$

$$
= \sum_{j=1}^{L} \mathbb{E}_{\boldsymbol{x} \sim \widehat{\mu}, \boldsymbol{y}_{0:j-1} \sim \pi_{\text{neg}}(\cdot|\boldsymbol{x})} \left[ \sum_{\boldsymbol{y}_j} \pi_{\text{neg}}(\boldsymbol{y}_j|\boldsymbol{x}, \boldsymbol{y}_{0:j-1}) \log \frac{\pi_{\text{neg}}(\boldsymbol{y}_j|\boldsymbol{x}, \boldsymbol{y}_{0:j-1}) \cdot \widehat{\mathbb{Z}}(\boldsymbol{x}, \boldsymbol{y}_{0:j})}{\pi_{\text{sft}}(\boldsymbol{y}_j|\boldsymbol{x}, \boldsymbol{y}_{0:j-1})} \right]
$$

$$
= \beta \cdot \sum_{j=1}^{L} \mathbb{E}_{\boldsymbol{x} \sim \widehat{\mu}, \boldsymbol{y}_{0:j-1} \sim \pi_{\text{neg}}(\cdot|\boldsymbol{x})} \left[ D_{\text{KL}} \left( \pi_{\text{neg}}(\cdot|\boldsymbol{x}, \boldsymbol{y}_{0:j-1}), \pi_{\text{sft}}(\cdot|\boldsymbol{x}, \boldsymbol{y}_{0:j-1}) \right) + \log \widehat{\mathbb{Z}}(\boldsymbol{x}, \boldsymbol{y}_{0:j}) \right]
$$

$$
\geq \beta \cdot \sum_{j=1}^{L} \mathbb{E}_{\boldsymbol{x} \sim \widehat{\mu}, \boldsymbol{y}_{0:j-1} \sim \pi_{\text{neg}}(\cdot|\boldsymbol{x})} \left[ \log \widehat{\mathbb{Z}}(\boldsymbol{x}, \boldsymbol{y}_{0:j}) \right].
$$

$$
= \beta \cdot \mathbb{E}_{\boldsymbol{x} \sim \widehat{\mu}, \boldsymbol{y}_{i,0:L} \sim \pi_{\text{neg}}(\cdot|\boldsymbol{x})} \left[ \sum_{j=1}^{L} \log \mathbb{Z}(\boldsymbol{x}, \boldsymbol{y}_{0:j}) \right].
$$

Now, we can prove the desired result by accounting for the gap in the success rate between the actual distribution over $\boldsymbol{x} \sim \mu$ and the empirical distribution induced by problems in the dataset $\widehat{\mu}$:

$$
J(\pi_{\text{neg}}) - J(\pi_{\text{sft}}) \geq \underbrace{J(\pi_{\text{neg}}) - \widehat{J}(\pi_{\text{neg}})}_{(a)} + \underbrace{\widehat{J}(\pi_{\text{neg}}) - \widehat{J}(\pi_{\text{sft}})}_{(b)} - \underbrace{J(\pi_{\text{sft}}) - \widehat{J}(\pi_{\text{sft}})}_{(c)}
$$

$$
\geq \beta \cdot \mathbb{E}_{\boldsymbol{x} \sim \widehat{\mu}, \boldsymbol{y}_{i,0:L} \sim \pi_{\text{neg}}(\cdot|\boldsymbol{x})} \left[ \sum_{j=1}^{L} \log \widehat{\mathbb{Z}}(\boldsymbol{x}, \boldsymbol{y}_{0:j}) \right] - \frac{c_0}{\sqrt{|\mathcal{D}_{\text{syn}}|}}
$$

$$
\geq \beta \cdot \mathbb{E}_{\boldsymbol{x} \sim \widehat{\mu}, \boldsymbol{y}_{i,0:L} \sim \pi_{\text{neg}}(\cdot|\boldsymbol{x})} \left[ \sum_{j=1}^{L} \log \mathbb{Z}(\boldsymbol{x}, \boldsymbol{y}_{0:j}) \right] - \frac{c_0}{\sqrt{|\mathcal{D}_{\text{syn}}|}} + \Delta,
$$

where $c_0$ is a constant that depends on the Rademacher complexity of the function class of policies $\pi_{\text{neg}}$ (coming from a uniform bound that we invokve to bound term *(a)*, since $\pi_{\text{neg}}$ depends on the dataset samples), and this term arises since the empirical distribution over prompts is not the same as the true population. This concentration term decays as the size of the synthetic data (number of problems) are scaled up. The term $\Delta$ denotes the overestimation error between the estimated advantages $\widehat{A}_{\tilde{\pi}}(\boldsymbol{x}, \boldsymbol{y}_{0:i-1}, \boldsymbol{y}_i)$ and the true advantages $A_{\tilde{\pi}}(\boldsymbol{x}, \boldsymbol{y}_{0:i-1}, \boldsymbol{y}_i)$, in expectation under the distribution of the learned policy. The estimation error $\Delta$ depends on $\pi_{\text{sft}}$ and the value of $K$ used if the rollout policy $\tilde{\pi}$ corresponds to the BoK($\pi_{\text{sft}}$) policy. This proves the theorem. $\square$

**Interpretation & perspectives.** Also note that the improvement in performance between $\pi_{\text{neg}}$ and $\pi_{\text{sft}}$ depends on the advantage estimate: if the advantage estimates are high, then this term is large, meaning that the more the fraction of high-advantage critical states, the higher the improvement. In addition, the bound also says that if the over-estimation $\Delta$ in the advantage estimate is large, the performance improvement is small. This is perhaps expected: consider the scenario when the BoK($\pi_{\text{sft}}$) policy is used to collect data, for a large $K$. In this scenario, the divergence between the empirical advantage estimate $\widehat{A}_{\tilde{\pi}}$ and the expected estimate $A_{\tilde{\pi}}$ is likely large. In the worst case, the estimate $\widehat{A}_{\tilde{\pi}}$ can arbitrarily overestimate $A_{\tilde{\pi}}$, as it would take on a high value even if there is just *one* sequence among the $K$ rollouts that successfully solves the problem. For example, a spurious step may be labeled incorrectly as critical in this case and training on negative data may not improve (consistent with running per-step DPO on an over-trained SFT checkpoint in Figure 8). On the other hand, if advantages are more accurate, training on negative data should improve performance.

## G  Additional Related Work

**Failure modes for supervised finetuning (SFT).** First, since SFT induces an open-loop [62] next-token prediction loss, prediction errors on even a single token can snowball during inference, leading

to poor performance on the prompts appearing in the dataitself [24, 45]. Second, even when an LLM has perfectly cloned the SFT data, it is prone to memorize "hard to learn" tokens [56], especially in planning and lookahead tasks [35, 36], which is critical for math reasoning. This leads to poor generalization [4, 15] and hallucination on new novel, test-tim prompts [26]. In this work, we study how synthetic data methods can address these failures via: (i) maximizing likelihood on positive data generated from both the SFT policy and a stronger teacher that enjoys improved coverage over new states, and (ii) preference optimization using the negative data generated from the SFT policy.

## H    Synthetic Data Generation

> **Prompt used for GSM8K/MATH synthetic data [29]**
>
> Please act as a professional math teacher. Your goal is to create high quality math problems to help students learn math. You will be given a math question. Please create a new question based on the Given Question and following instructions.
> To achieve the goal, you have one job.
> # Please generate a similar but new question according to the Given Question.
> You have four principles to do this. # Ensure the new question only asks for one thing, be reasonable, be based on the Given Question, and can be answered with only a number(float or integer). For example, DO NOT ask, 'what is the amount of A, B and C?'.
> # Ensure the new question is in line with common sense of life. For example, the amount someone has or pays must be a positive number, and the number of people must be an integer.
> # Ensure your student can answer the new question without the given question. If you want to use some numbers, conditions or background in the given question, please restate them to ensure no information is omitted in your new question.
> # You only need to create the new question. Please DO NOT solve it.
> Given Question: <insert question from original dataset here>
> Your output should be in the following format:
> CREATED QUESTION: <your created question>

For GSM8K, we replace the phrase "Your goal is to create high quality math problems to help students learn math." with "Your goal is to create high quality math *word* problems to help students learn math.", as we found this to produce problems that were closer to GSM-style problems.

To generate the synthetic data, we used OpenAI credits worth approximately 3000 US dollars.

## I    Details on Star Graph Problem

The star graph problem we study is borrowed from  Bachmann and Nagarajan [4], where given a graph in the shape of a star and a query (center/end node pair), the model is asked to output the full path between the start/end nodes.

**Goal.** Bachmann and Nagarajan [4] show that $\pi_{\mathrm{sft}}$ minimizes SFT loss by memorizing the "hard-to-predict" node adjacent to the center, and copying the rest of the path from the input graph. This task is highlights the failure of SFT at planning problems (akin to math reasoning). Thus, we use this as a case study to understand:

- when accurate advantage estimation is possible with few negative samples from the $\pi_{\mathrm{sft}}$ model.

- whether there are generalization benefits of advantage-weighted RL when advantage estimates are accurate

- when advantage-weighted RL can unlearn the memorized feature that causes $\pi_{\mathrm{sft}}$ to fail.

**SFT dataset.** The data we use for supervised fine-tuning consists of 30000 of random star graphs (see examples below) where each graph has a centre node with out degree 2. Hence, there are two paths that originate from the centre node. Each path from the center to one of the end nodes is of length 4. Each node on the path is denoted with a randomly sampled number from 0 to 20. For example, in the sample "8,3|3,10|14,13|10,1|17,14|8,17/8,13=8,17,14,13". The graph is given by the

adjacency list "8,3|3,10|14,13|10,1|17,14|8,17/8,13", the query is denoted by "8,13", and the correct path is given by "8,17,14,13".

**Test-time inference from the model.** At test time, the input to the LLM is only thw graph and the query: "8,3|3,10|14,13|10,1|17,14|8,17/8,13=" and the model is expected to generate the full path from start node 8 to end node 13. When evaluating the test performance of an LLM, we calculate $0/1$ accuracy averaged over 1000 test star graphs (that are different from train star graphs). The accuracy on a sample is 1 when the LLM accurately predicts all nodes in the graph.

**Failure models of the SFT model, $\pi_{\text{sft}}$.** A model with perfect accuracy (0 error) would be the one that has accurately learned the correct feature of backtracking the path from the end node to the start node, and then producing it in reverse. This computation is precisely what makes the adjacent token "hard-to-fit". On the other hand, if the LLM minimizes next-token prediction loss during SFT by instead memorizing the hard-to-fit adjacent token by overfitting on the random input graph instance, at test time the accuracy would be zero. An intermediate solution that SFT model instead learns is to output a path that is adjacent to the node. At training time, it only needs to memorize which of the two possible path to predict. Note that even this solution does not require the model to backtrack, and is thus easier to quickly learn with a few samples. This would quickly minimize the loss on all but the adjacent node, which the model memorizes as training progresses. On the test set, this model would then have $50\%$ test accuracy. Note, that as we increase the size of the graph or the node vocabulary size it becomes easier for the model to overfit on the hard to predict adjacent token given random combinations of the input graph. Thus, we choose the vocabulary size to be 20, which is higher than what is needed to represent any input graph of this size.

Below we provide examples from degree two, path length 4, node 20 problem, where

---

**Examples of 20 node path length 4 star graph problem**

Example 1: 8,3|3,10|14,13|10,1|17,14|8,17/8,13=8,17,14,13
Example 2: 14,16|8,10|9,5|3,14|9,3|5,8/9,16=9,3,14,16
Example 3: 14,1|10,4|9,7|10,17|4,9|17,14/10,7=10,4,9,7
Example 4: 19,8|7,18|14,15|15,7|14,19|8,10/14,10=14,19,8,10
Example 5: 1,6|10,1|6,12|10,17|17,18|18,5/10,12=10,1,6,12

---

**SFT Training details.** We finetune a pretrained GPT-2 model with 125 million parameters. We train with a batch size of 128, Adam without any weight decay, and a constant learning rate of $1e-5$ .

**Advantage estimation and per-step DPO training equivalent to advantage-weighted RL.** For a sample from $\pi_{\text{sft}}$, we estimate the advantage of each step by sampling 5 rollouts conditioned on the subsequence uptill the step. We then pair subsequences with shared prefix, $\boldsymbol{y}_{1:i}$ differing in the last step $+\boldsymbol{y}_{i+1}$ vs. $-\boldsymbol{y}_{i+1}$, where the former is the one with the highest estimated advantage and the latter is the one with the lowest estimated advantage. Note that this preference pair construction, closely approximates the preference pair distribution in Theorem 6.1, which would imply that the DPO objective being optimized closely approximates advantage weighred RL in Equation 4.

Given these pairs for a batch of star graph problems in SFT data, we update the model with a single gradient step on the DPO objective in Equation 1. In the next iteration, advantage is estimated and pairs are constructed on a fresh batch of star graphs. We set $\beta = 0.1$ in the DPO objective and use the same batch size (one preference pair per star graph). Starting from an SFT checkpoint we train in the above manner for at least 200 iterations. The SFT model is trained for over 600 iterations.

## J  Implementation Details

**Datasets and pretrained LLMs.** We run all our experiments on GSM8K and MATH datasets. Each dataset has about 7.5k training examples. The GSM8K has about 1.3k and MATH has 5k test examples. We conduct experiments with DeepSeek-Math-7B pretrained LLM and LLama2-7B, both of which have pretrained weights publicly available on Huggingface.

**Details for SFT/RFT training.** For our positive data scaling results, the SFT model is trained for 5 epochs with a learning rate of $1e-5$, and a batch size of 64 for all sizes of $\mathcal{D}_{\text{syn}}$. We use a holdout validation set to choose the checkpoint and report the performance of the best performing checkpoint across the five epochs. To generate RFT data we only train the SFT model for 2 epochs (under-trained checkpoint). For each question we sample $M = 100$ times, with a temperature of $0.7$ and following

Yuan et al. [67] we retain at most 4 most diverse (based on edit distance) and correct completions. This is for our results in Figure 2(a,b). For Figure 2(c), we sample more than 4 correct solutions and keep sampling responses until we have a dataset of size 10k, 20k, ..., 50k, when questions are given by the $\mathcal{D}_{\text{syn}}$ of size 8k and 16k. For our experiment on the RFT dataset with purposely inserted spurious steps, as we describe in the main paper, we obtain spurious steps by computing which intermediate steps in a negative response lead to most incorrect solutions and randomly insert this in between reasoning steps for a positive solution in RFT data. See examples below.

**Details for per-step DPO training.** Training data for DPO is generated in the procedure outlined in [23]. The value of $K$ is 5 for the BoK($\pi_{\text{sft}}$) policy used to estimate the advantage, and a single rollout from the best-of-K policy is used to estimate the advantage of each step in "rejected" response, where the "chosen" and "rejected" pairs for each question are generated from the RFT data, using the method in [38]. We train per-step DPO with $\beta = 0.3$ for all $\mathcal{D}_{\text{syn}}$ from GSM8K, and $\beta = 0.1$ for all $\mathcal{D}_{\text{syn}}$ from MATH. We train for three epochs, with a learning rate of $1e - 6$ and a batch size of 64 prompts. We use the rmsprop optimizer and a cosine learning rate decay with a warmpup ratio of 0.1.

Overall, to replicate our experiments, we need approximately 1000 hours of compute on H100 clusters with 8 GPUs.

---

**RFT data with spurious steps highlighted**

**Prompt:**
Karen's work tote bag is twice the weight of her husband Kevin's briefcase when the briefcase is empty. When Kevin puts his laptop and work papers in his briefcase, it is twice the weight of Karen's tote. Kevin's work papers are a sixth of the weight of the contents of his full briefcase. If Karen's tote weighs 8 pounds, how many more pounds does Kevin's laptop weigh than Karen's tote?

**Response:**
If Karen's tote weighs 8 pounds, and it is twice the weight of Kevin's empty briefcase, then Kevin's empty briefcase weighs 8/2 = «8/2=4»4 pounds. In words, "Karen's tote weighs 8 pounds" is "T = 8", so thus Keven is 2 pounds. When Kevin puts his laptop and work papers in his briefcase, it is twice the weight of Karen's tote at 8 pounds, so the briefcase now weighs 8+4 = «8+4=12»12 pounds. Kevin's work papers are a sixth of the weight of the contents of his full briefcase, which weighs 12 pounds, so the work papers weigh 12/6 = «12/6=2»2 pounds. If Kevin's laptop and work papers weigh 12 pounds in total, and the work papers weigh 2 pounds, then the laptop weighs 12-2 = «12-2=10»10 pounds. Thus, Kevin's laptop is 10-8 = «10-8=2»2 pounds heavier than Karen's tote. The answer is 2.

---

**RFT data with spurious steps highlighted**

**Prompt:**
Carl has a jar full of marbles. He takes out 12 marbles to play a game with, but he accidentally drops them and 1/2 the marbles get lost. So Carl takes out 10 more marbles. While he is playing his game his mother comes home with another bag of marbles for him, which has 25 marbles in it. If Carl can't find his lost marbles, then how many marbles is he going to put in the jar after he plays his game, from both his original marbles and the new ones?

**Response:**
First find out how many marbles Carl loses: 12 marbles * 1/2 = «12*1/2=6»6 marbles Then add that number to the number of marbles Carl takes out of the jar to play his game to find out how many total marbles he takes out: 6 marbles + 10 marbles = «6+10=16»16 marble. Now add that number to the number of marbles in the jar to find out how many marbles are left: 16 marbles + 12 marbles = «16+12=28»28 marbles. Then add in the marbles Carl gets from his mother to find out how many marbles he has in total: 16 marbles + 25 marbles = «16+25=41»41 marbles. The answer is 41.

---

## K   Negative Data Identifies Spurious Steps with Advantage Estimates

In this section, we present several examples from the GSM8K and MATH synthetic datasets of model-generated (from $\pi_{\text{sft}}$) positive data with spurious steps. Then, we show how computing

advantage values accurately identify the spurious steps. We highlight the steps that were deemed to be low or high advantage based on the estimated advantage value. The darker the intensity of the color, the lower the advantage of that step. With this qualitative probe, we see that the steps that are darkest are indeed either incorrect or correct but involve a reasoning step that is irrelevant for solving the problem at hand.

---

**Spurious step identified by negative data**

**Question:** Ken created a care package to send to his brother, who was away at boarding school. Ken placed a box on a scale, and then he poured into the box enough jelly beans to bring the weight to 2 pounds. Then, he added enough brownies to cause the weight to triple. Next, he added another 2 pounds of jelly beans. And finally, he added enough gummy worms to double the weight once again. What was the final weight of the box of goodies, in pounds?

**Positive response with identified spurious step:**
When the weight was originally on the scale, the box weighed 0 pounds. After Ken added the jelly beans to the box, its weight was 2 pounds - 0 pounds = «2-0=2»2 pounds. Then Ken added the brownies to the box, which caused the box to weigh 2 pounds * 3 = «2*3=6»6 pounds. The next step was to add 2 pounds of jelly beans, which caused the box to weigh 6 pounds + 2 pounds = «6+2=8»8 pounds. First, we need to figure out what the weight of the box was when Ken put the jelly beans in it. 2 pounds of jelly beans was enough to triple the weight, so the box was at 2 pounds / 3 = «2/3=.6666666666666667».6666666666666667 pounds. Finally, Ken added the gummy worms, causing the box to weigh 8 pounds * 2 = «8*2=16»16 pounds. The answer is 16

---

**Spurious step identified by negative data**

**Question:** Betty is saving money for a new wallet which costs $100. Betty has only half of the money she needs. Her parents decided to give her $15 for that purpose, and her grandparents twice as much as her parents. How much more money does Betty need to buy the wallet?

**Positive response with identified spurious step:**
Betty's parents gave her 15, so her grandparents gave her 2 * 15 = «15*2=30»30. In total, Betty received 30 + 15 = «30+15=45»45. The total amount of money Betty needs is 100 * 2 = «100*2=200»200. Betty needs 100 / 2 = «100/2=50»50 in total. Betty still needs to save 50 - 45 = «50-45=5»5. The answer is 5

---

**Spurious step identified by negative data**

**Question:** To make pizza, together with other ingredients, Kimber needs 10 cups of water, 16 cups of flour, and 1/2 times as many teaspoons of salt as the number of cups of flour. Calculate the combined total number of cups of water, flour, and teaspoons of salt that she needs to make the pizza.

**Positive response with identified spurious step:**
Since 1 cup is equal to 16 teaspoons, then Kimber needs 128/16 = «128/16=8»8 cups of salt. So in total, she needs 10+16+8 = «10+16+8=34»34 cups of water, flour, and salt. In cups, the amount of salt is 128/6 = «128/6=21.33»21.33 cups The answer is 34

**Question:** Eight friends compete in a pie eating contest. The chart displays the number of pies each person consumed. Emily ate the most pies, and Mike ate the least. How many more pies did Emily eat compared to Mike?

```
[asy]\ndefaultpen(linewidth(1pt)+fontsize(10pt));
\npair[] yaxis = new pair[10];\nfor( int i = 0 ; i < 10 ; ++i )
{\n\nyaxis[i] = (0,i);\n\ndraw(yaxis[i]--yaxis[i]+(17,0));\n}
\ndraw((0,0)--(0,9));\ndraw((17,9)--(17,0));
\n\nfill((1,0)--(1,5)--(2,5)--(2,0)--cycle,grey);
\nfill((3,0)--(3,8)--(4,8)--(4,0)--cycle,grey);
\nfill((5,0)--(5,3)--(6,3)--(6,0)--cycle,grey);
\n\nlabel("0",yaxis[0],W);
\nlabel("1",yaxis[1],W);\nlabel("2",yaxis[2],W);
\nlabel("3",yaxis[3],W);\nlabel("4",yaxis[4],W);
\nlabel("5",yaxis[5],W);\nlabel("6",yaxis[6],W);
\nlabel("7",yaxis[7],W);\nlabel("8",yaxis[8],W);
\nlabel("Friends/Participants",(8.5,0),S);
\nlabel("Results of a Pie Eating Contest",(8.5,9),N);
\nlabel(rotate(90)*"$\\#$ of Pies Eaten",(-1,4.5),W);[/asy]
```

**Positive response with identified spurious step:**
To solve this problem, we need to identify the number of pies Emily and Mike consumed, and then find the difference between their consumption. From the chart provided: - Emily: Consumed 8 pies. - Mike: Consumed 1 pie. To determine how many more pies Emily ate compared to Mike, we first identify how many pies each person ate. - Emily ate 5 pies. - Mike ate 1 pie. Now, calculating the difference between Emily's and Mike's consumption: [ Difference = Emily's consumption - Mike's consumption = 8 - 1 = 7. ] Therefore, Emily ate 7 more pies than Mike. The answer is 7

---

**Question:** When the base-16 number $1AB3_{16}$ is written in base 2, how many base-2 digits (bits) does it have?

**Positive response with identified spurious step:**
To solve this problem, we first need to convert the base-16 number $1AB3_{16}$ to base-2. Step 1: Convert each digit of the number from base-16 to base-2. The number $1AB3_{16}$ consists of the digits 1, A, B, and 3. Step 2: Convert each digit individually to base-2 representation. - $1_{16} = 0001_2$ - $A_{16} = 1010_2$ - $B_{16} = 1011_2$ - $3_{16} = 0011_2$ Step 3: Combine the base-2 representations of each digit. Concatenating the individual base-2 representations, we get: $1AB3_{16} = 0001101010110011_2$ Step 4: Count the total number of digits in the base-2 representation. The base-2 representation of $1AB3_{16}$ is: $0001101010110011_2$ The base-16 number $(1AB3_{16})$ can be written in base 10 (decimal) as: [ $1 \times 16^3 + A \times 16^2 + B \times 16 + 3$ ] where $(A)$ and $(B)$ are decimal digits. This sequence of digits has a total of 16 digits. The answer is boxed{16}

