# OpenReview forum: "RL on Incorrect Synthetic Data Scales the Efficiency of LLM Math Reasoning by Eight-Fold"
_NeurIPS.cc/2024/Conference — NeurIPS 2024 poster_

### Official Review · Reviewer_iuJa · 2024-07-09

**Soundness:** 3
**Presentation:** 3
**Contribution:** 4
**Rating:** 8
**Confidence:** 3

**Summary:**

There have been previous attempts at doing finetuning on positive synthetic examples to improve LM reasoning, but the performance gain from these attempts are generally quite limited (and possibly even negative). In this paper, the authors take a different approach which also accounts for negative examples and “critical” intermediate steps, improving upon previous approaches.

**Strengths:**

I was really impressed by this paper – at least for me, it contains quite a lot of insights, and is substantially better than most papers on synthetic data that I’ve seen in the past.
- I thought the idea of using negative data was fairly natural but also good, and I’m glad to see it implemented in practice
- I found that the proposed link to model collapse seemed really interesting, which allegedly can occur if a model generating synthetic data has memorized the training data using a spurious feature.
- I really liked the idea of looking at “critical steps”, and I thought the approach of relating this to credit assignment was impressive.

More generally, I liked that the authors really made clear attempts to both quantify and explain their empirical observations. For example, I appreciated the section comparing the sample efficiency of self-generated and GPT-4/Gemini 1.5 Pro synthetic data, which does both.

**Weaknesses:**

The biggest concrete thing that I felt was missing was discussion about computational costs. I think this is really important because I’d be interested in the scalability of approaches like the ones outlined in the paper. I think having some comparison of the overall FLOP required for finetuning (including the FLOP for synthetic data generation) would be very helpful.

Another weakness is that in my understanding, the main approaches in the paper apply most directly to mathematics, since identifying critical steps involves checking results based on MC rollouts. There’s still a question about whether or not similar approaches can be applied to LMs in domains where outputs are harder to “verify” – though to be clear, I think it’s entirely reasonable that the authors focus on math in the context of this paper.

Lines 197-199: This compares the scaling exponent for data for the MATH and GSM8K datasets with the exponent from the Chinchilla paper (on MassiveText). How do we know that this implies improving coverage over samples? Other possible reasons for the different exponent come to mind – e.g. different functional forms for the scaling law in finetuning and pretraining, use of different datasets, etc.

As a minor comment, I think “It is predicted that we will run out of high-quality internet data by 2026” should be modified to be about human-generated public text data in particular, and it looks like the median year from source [53] is 2028 rather than 2026.

**Questions:**

- Could the authors please provide some information about the computational costs of their approach? (perhaps comparing this for SFT, RFT, and per-step DPO)
- Clarification for lines 197-199: How do we know that this implies improving coverage over samples, rather than the other reasons I mentioned?
- Clarification for lines 203-204: Is it correct to interpret this as saying that “around 4% of sampled outputs from the SFT model are used for RFT”?

**Limitations:**

I think the authors did a fair job discussing the limitations of their work. One minor weakness is that the paper is framed around when synthetic data improves LM reasoning, but the investigation itself is more narrowly focused on mathematics.

---

> ### Author Rebuttal · Authors · 2024-08-07
>
> Thanks for the feedback and for the positive assessment of our paper! We are grateful to you for the comments and kind words regarding the contribution of our paper. To address the questions, we have added a discussion of computational costs below and answer the other questions raised. **Please let us know if your questions are addressed, and if so, we would be grateful if you might be willing to increase the score further.**
>
> ___
>
> ## Scalability of approaches; comparison of the overall FLOP required for fine-tuning
>
> Thanks for the question! We will add a more formalized version of the discussion below to the final version of the paper. To understand the scalability of SFT, RFT, and per-step DPO, we make the following comparisons. First, SFT exclusively uses synthetic prompts and responses generated from more capable models. Assuming inference FLOPs are $2 \times N \times D$, this cost scales with the parameter size ($N$) of the more capable model used to generate the data for every new synthetic question, whereas RFT and per-step DPO only require running more rollouts with a much smaller 7B model for fewer questions, not incurring the inference FLOPs of the more capable model. Thus in a FLOPs-matched comparison, RFT / per-step DPO should have an even bigger edge over SFT.
>
> Now we perform a comparison of FLOPs / compute costs for  RFT and per-step DPO, for a fixed number of synthetic prompts. We can break the total FLOPs into two parts: **(a)** inference FLOPs needed to generate data from the SFT policy for running RFT and per-step DPO, and **(b)** training FLOPs for RFT and per-step DPO.
>
> **Regarding (b),** we train both RFT and per-step DPO for an equal number of steps, with an identical number of forward and backward passes through the model: more precisely, since the DPO loss utilizes two samples together, we run DPO with half the batch size to fit on the GPU memory. Put together, this should lead to an equal number of forward and backward passes for per-step DPO and RFT. The training FLOPs are typically given by $6 \times N \times D$, which should be the same for both RFT and per-step DPO.
>
> **Regarding (a),** we compare the number of samples that need to be drawn from the SFT policy for both RFT and per-step DPO. For RFT, to collect enough positives from $\pi_\mathrm{sft}$, we draw 100 samples per question and filter for positive ones. For DPO, if the accuracy of $\pi_\mathrm{sft}$ is $p$, then with high probability, identifying a single positive and negative sample for a prompt takes $\approx max(1/p, 1/(1-p))$ samples. Now, for computing advantage estimates of each step in the negative response, we set the maximum number of steps per generation as 10 and sample 5 MC rollouts conditioned on each step. Thus, in total per-step DPO requires $\approx max(1/p, 1/(1-p)) + 50$ samples per question, which is a value smaller than $100$ for RFT, when we plug in $p \approx  0.4$ and $p \approx 0.7$ for MATH and GSM8K respectively (this is the accuracy of the SFT model in our experiments). **Thus, DPO is more computationally efficient than RFT in our experiments.**
>
> ___
> ## Clarification for lines 197-199: How do we know that this implies improving coverage over samples, rather than the other reasons I mentioned?
>
> Thanks for bringing this up! Indeed the difference in scaling law exponents can come from a different functional form of the scaling law. We originally wrote this statement because the approx. exponent of the scaling law of Llama2-70B on GSM8K (derived from training on 1, ½, ¼ subsets of only real GSM8K training data) is -0.3 (following the result from Yuan et al. [68]) and -0.105 for DeepSeek-7B on MATH (see below for the result), whereas we observed -0.15 with synthetic data on GSM8K and -0.05 on MATH, which is substantially lower. Since the only difference is in the datasets (at least for MATH), we attributed this to the coverage of samples. That said, you are right that a comparison with pre-training scaling laws may not be correct here, so we will update this line to note the difference against training models on real math data, rather than contrasting it with pre-training scaling laws.
>
> | Training data fraction |   1   |  1/2  |  1/4  |  1/8  |  1/16 |  1/32 |
> |:----------------------:|:-----:|:-----:|:-----:|:-----:|:-----:|:-----:|
> |       Test error       | 0.624 | 0.671 | 0.742 | 0.788 | 0.832 | 0.905 |
>
> ___
> ##  The main approaches in the paper apply most directly to mathematics, since identifying critical steps involves checking results based on MC rollouts.
>
> We only study our approach on MATH and GSM8K due to computational constraints associated with running it on other domains, and are happy to change the title of the paper to include “math reasoning” instead of “reasoning” to clearly scope the contributions of the paper, if the reviewer thinks that would be more appropriate. That said, we do think that identifying critical steps using MC rollouts and per-step RL can also be used for coding problems, where each line or each function call can correspond to a step (e.g., see StepCoder (Ma et al. 2023), which runs a similar style of MC rollouts to collect data for training process reward models on coding problems). We will add a discussion of this in the paper.
>
> ___
> ## Clarification for lines 203-204:
>
> Yes, your interpretation is correct in that only 4 out of the 100 sampled responses are used for training with RFT, following the procedure in Yuan et al. 2023.
>
> ___
> ## Typo / minor concern: “It is predicted that we will run out of high-quality internet data by 2026
>
> Thanks for catching the typo! We will update the statement to reflect 2028, the median year in that reference and use the term “human-generated public text data” instead of internet data to address the concern.

---

> > ### Comment · Reviewer_iuJa · 2024-08-11
> >
> > Thanks for the really comprehensive response! I'll keep my official score at 8 - while I think the work is excellent, the description for a score of 9 requires the work to be "groundbreaking", which I think is a very high bar. However, I should emphasise that my main concerns have been addressed, and if it were possible I'd raise my score by 0.5 points to reflect this.

---

> > > ### Author Response · Authors · 2024-08-12
> > > **Thank you for your appreciation**
> > >
> > > Thank you immensely for your kind appreciation and very positive assessment of our work! To further improve our paper, we will incorporate the clarifications and discussion we provide in our responses.
> > >
> > > -Authors

---

### Official Review · Reviewer_8A1v · 2024-07-12

**Soundness:** 3
**Presentation:** 3
**Contribution:** 3
**Rating:** 6
**Confidence:** 4

**Summary:**

The paper investigates how to effectively leverage synthetic data to improve reasoning capability in the GSM8K and MATH datasets. The authors identify that sampling correct synthetic data from a fine-tuned model is more sample-efficient but comes with the risk of overfitting artifacts in the synthetic data. Instead, they propose a per-step DPO to leverage negative synthetic data to enhance sample efficiency.

**Strengths:**

1. Synthetic data for reasoning is a highly important topic.
2. The idea is straightforward and effective.
3. The experiments are extensive, with comprehensive ablation studies.

**Weaknesses:**

1. Missing Relevant References: The paper lacks references to relevant literature. Scaling laws on how synthetic data plateaus or performs slower than real data during pretraining have been observed and analyzed in several studies [1][2]. Additionally, the observation that self-generated data is more sample-efficient has been noted in research on images and machine translation [3][4][5].
2. Unclear Per-Step DPO Definition: The per-step DPO is not fully defined. In Line 180, how is the “first pit” $\hat{y_c}$ determined? Does it involve looping over all intermediate steps from the beginning to find the smallest $c$? It would be helpful for the authors to provide an algorithmic outline in the appendix. In Theorem 6.1, the per-step DPO is trained with pairs (x, $[y_{1:i}, +y_{i+1}]$, $[y_{1:i}, -y_{i+1}]$), whereas Line 180 defines it as using $+y$ instead of $[y_{1:i}, +y_{i+1}]$. Which version is implemented in the experiments, and why?
3. Claims on Sample Efficiency: The improved performance on filtered $\mathcal{D}_{syn}$ from the fine-tuned model requires sampling 100 samples per question, and the per-step DPO samples require Monte Carlo rollouts for intermediate steps. This adds significant computational overhead during inference, much more than during the fine-tuning stage. A proper comparison of sample efficiency should include the inference compute spent generating these examples. Given the same computational resources, would DPO perform better as it does not require identifying the “first pit” while utilizing more synthetic data?
4. Explanation of Figure 7: The explanation of Figure 7 is unclear. How are the average Q-values calculated? What if a problem requires fewer than 8 steps in GSM8K? Will the average for step 8 be biased toward harder questions?

### Reference
[1] Dohmatob, Elvis, et al. "A tale of tails: Model collapse as a change of scaling laws." arXiv preprint arXiv:2402.07043 (2024).

[2] Fan, Lijie, et al. "Scaling laws of synthetic images for model training... for now." Proceedings of the IEEE/CVF Conference on Computer Vision and Pattern Recognition. 2024.

[3] Mobahi, Hossein, Mehrdad Farajtabar, and Peter Bartlett. "Self-distillation amplifies regularization in hilbert space." Advances in Neural Information Processing Systems 33 (2020): 3351-3361.

[4] Gu, Jiatao, et al. "Non-autoregressive neural machine translation." arXiv preprint arXiv:1711.02281 (2017).

[5] Zhou, Chunting, Graham Neubig, and Jiatao Gu. "Understanding knowledge distillation in non-autoregressive machine translation." arXiv preprint arXiv:1911.02727 (2019).

**Questions:**

1. Line 138: Why can we assume that the answers generated via this process are accurate? Is the model prompted with the correct solution?
2. Line 166: The sentence is not well-written.
3. DPO Suitability for Math: DPO is derived from the Bradley-Terry model, where preference follows a probabilistic distribution with softmax over reward values. For mathematical generation, however, correct answers should always be preferred over incorrect ones with probability 1. The framework, especially naive DPO, seems unsuitable for the math set, which might explain its weak performance. The KTO method, cited in the paper, proposes a different RLHF method that considers positive and negative distributions directly. How would the KTO method perform? Would the per-step version of KTO also outperform naive KTO?
4. Line 302: Why are the advantage values always non-positive?
5. Data Generation Protocols: It would be more self-contained to include critical data generation protocols in the paper rather than referencing “Training data for DPO is generated in the procedure outlined in [23]” in Appendix H at line 800.

The reviewer will consider increase the score if some of the weakness and questions are addressed.

**Limitations:**

Limitations are discussed.

---

> ### Author Rebuttal · Authors · 2024-08-07
>
> Thank you for the feedback. To address the concerns, we add new results applying per-step RL to KTO and RFT as well, and compare computational costs. We also clarify the per-step DPO algorithm, advantage functions and Fig 7. **Please let us know if these responses address your concerns, and if so, we would be grateful if you are willing to raise your score.**
>
> ___
> ## [New exp.] Per-step DPO definition, KTO results (W2, Q3)
>
> As we understand, there are two concerns:
> - (1) does per-step DPO work well because DPO is a weak baseline, and would the benefits of per-step RL translate to other RL methods?;
> - (2) the exact formulation of per-step DPO.
>
> To answer (1), we show that per-step variants of other RL methods also improve over them. We add new results extending RFT and KTO to their corresponding per-step variants using both positive and negative data. We find that their respective per-step variants outperform both RFT (Fig 1 in PDF) and KTO (Fig 4 in PDF) on both GSM8K and MATH. For RFT, the per-step variant exactly optimizes a sampled estimate of Thm 6.1.
>
> In Fig 4 (in PDF), we find that while KTO is slightly worse than DPO, the per-step version of KTO consistently outperforms KTO. That said, we still emphasize that **our point is not to propose per-step DPO as an algorithm, but to show that our conceptual model of per-step advantages and negative data can enhance the efficiency of synthetic data.** We chose DPO as a base algorithm, but this is not the only choice.
>
> Regarding (2), we clarify that our implementation of per-step DPO is the one in L180. To avoid confusion, we present it as an algorithm box (Panel 5 in 1-page PDF). While this does deviate a little from Thm 6.1, it allowed us to build on Hwang et al. 2024’s codebase while still conforming to our conceptual model. In our implementation, we generated preference pairs using only negative responses. When we also consider low-advantage steps in positive data to construct preference pairs (closer to formulation in Thm 6.1), we see a further improvement of 1.5-2x (Panel 5). We also note that **our formulation from Thm 6.1 has been successfully utilized by [a paper that appeared on arXiv after NeurIPS deadline](https://arxiv.org/pdf/2406.18629), confirming the efficacy of our framework as well.**
>
> ___
> ## Claims on sample efficiency, cost of per-step DPO vs RFT (W3)
>
> To clarify, when we say per-step DPO is more efficient than RFT/SFT, we mean that per-step DPO achieves better accuracy _for the same number of synthetic prompts_, vs SFT or RFT. We consider only the number of synthetic prompts here as these require querying proprietary models, which costs inference FLOPs, and hence is more expensive than sampling the SFT model.
>
> However, for a fixed set of synthetic problems, we can compare the efficiency of per-step DPO & RFT, in terms of the number of samples from the SFT policy used. For RFT, to collect enough positives from SFT, we draw 100 samples per prompt & filter correct ones. For DPO, if the accuracy of SFT is $p$, then with high probability, identifying *one* positive and negative sample takes $max(1/p, 1/(1-p))$ samples / prompt. For computing advantages in the negative response, we set the maximum number of steps per generation as 10 and sample 5 MC rollouts from each step. Thus, in total per-step DPO requires $max(1/p, 1/(1-p)) + 5* 10$ samples per prompt, which is smaller than $100$ for RFT, when we plug in $p \approx  0.4$ and $p \approx 0.7$ for MATH and GSM8K respectively. **Thus, per-step DPO is more computationally efficient than RFT in our results.**
>
> **Please see global response for more details.**
> ___
> ## Explanation of Fig. 7
>
> In Fig. 7, our main claim is that the per-step approach of using positive and negative data uniformly improves Q-values across all steps compared to the SFT policy. This should be expected since advantage-weighted RL optimizes the RL expected reward objective. On the other hand, standard DPO which does not maximize the per-step RL objective does not improve Q-values over that of the SFT policy, at earlier steps.
>
> The avg. Q-value at step $T$ is averaged over only those responses with $\geq T$ steps. We agree that later steps may only appear in harder problems & may have lower Q-values, but our goal is to compare per-step RL with the SFT policy (gray dashed line) *for the same step* and not to compare Q-values across different steps.
>
> ___
> ## Why are advantages non-positive?
>
> In RL, advantages under the optimal policy are non-negative since the advantage is given by the difference between Q-value for an action and the maximum of Q-value at that state. In contrast, our definition in Eq. 3 relies on the choice of $\tilde{\pi}$, so you are correct that Eq. 3 may not always be non-positive for arbitrary $\tilde{\pi}$. However, when setting $\tilde{\pi} = \mathrm{BoK}(\pi_\mathrm{sft})$, and when $K$ is large, w.h.p. we would expect Eq. 3 to be non-positive since $\tilde{\pi}$ should improve over the choice of step $\hat{y}_{i-1}$ prescribed by the SFT policy. This may not be true when $K$ is small or when sampling error is present, and we will clarify this in the paper.
>
> ___
> ## Clarifications about Line 138.
>
> In general, we cannot guarantee that generated data will be correct, but we follow the recipe of Li et al. 2024 that asks the model to also verify its responses. Li et al. 2024, Liu et al. 2023 follow the same data protocol and show that scaling up synthetic data this way can improve 7B models to beyond GPT4 performance, indicating that data quality may not be as bad.
>
> ___
> ## Missing references.
>
> Thanks for pointing out these references, we will cite & discuss them. We believe our contributions go beyond these works: for e.g., while [1, 2] find slower convergence with synthetic data, and [3, 4, 5] discuss efficiency of self-gen. data, neither of them study how to use negative data or propose a conceptual framework to understand it. None of these focus on reasoning with modern LLMs.

---

> ### Comment · Reviewer_8A1v · 2024-08-11
>
> I appreciate the response and the new experiments on KTO. However, I am not entirely convinced by the claim that "*per-step DPO is more computationally efficient than RFT in our results*" as the author is drawing 100 samples per prompt, which introduces the possibility of variance when drawing 10 samples per prompt while creating more prompts in RFT. This variant could potentially surpass the original RFT with 100 samples per prompt. The new experiments on the overfitting of RFT, however, demonstrate that given abundant compute resources, per-step DPO clearly outperforms. Consequently, I have increased my score.

---

> ### Author Response · Authors · 2024-08-12
> **Clarification on computational costs of per-step DPO and RFT**
>
> Thank you for your response and for increasing the score!
>
> We are not sure if we fully follow your comment on the “computational efficiency” of RFT vs. per-step DPO, and might be misunderstanding your proposed variant of RFT, and the precise notion of “variance” being referred to. Therefore we request you to kindly correct us if the following response does not address your comment. **Please let us know.**
>
> _We begin by belaboring that the goal of our work is not to propose “per-step DPO” as novel algorithm_, but more to highlight our conceptual framework of positive and negative synthetic data, where credit assignment with negative data can address issues caused by overfitting on spurious steps in positive synthetic data. This general point is also evidenced in our other comparisons: per-step RFT is better than RFT, and per-step KTO is better than KTO.
>
> **Sampling new prompts is substantially more costly in our setting than sampling new responses.** We apologize if this was not clear but our claim “per-step DPO is more computationally efficient than RFT in our results” is only meant to apply to the setting with the **same set of synthetic prompts**. This is because obtaining new synthetic prompts requires querying more-capable models (like GPT-4, Gemini 1.5 Pro) in all of our experiments, which would incur a much higher factor times the computational cost of the 7B models that we study (these models are more capable than Llama2-70B, so it is safe to say that their sizes are at least 10x larger, which means at least 10x the computational cost for sampling more questions).
>
> While we agree that in some uses cases where sampling new prompts using a 7B model might be sufficient, there can be variants of RFT with more prompts (and less solutions per prompt) which could be more computationally efficient, this is not true in our setting, and we are not aware of any work that samples synthetic math questions (of the same quality as GPT/Gemini) with smaller scale models.
>
> ___
>
> **Empirical justification for the cost comparison given same set of prompts:** Within the context of the same set of prompts, more samples for RFT (e.g., 100) should reduce variance, it might lead to overfitting as we have shown in Section 5.
> Our results in Figure 1c (1-page PDF) compares the best configuration of RFT (which uses 4 diverse and correct samples out of the 100 samples) with per-step DPO. Note that these 4 out of 100 samples are selected based on the techniques prescribed by Yuan et al. [68] to obtain the most competitive RFT performance. For comparison, in our experiments on the 8k sized synthetic data which has 8k prompts, we got best performance from RFT by sampling 100 responses per prompt (total of **8k * 100 = 800k**) and selecting 4 diverse and correct responses per prompt from them. On the other hand, for per-step DPO we only need one positive and one negative trace. Even accounting for the advantage estimation run on every step in the negative trace, we only needed to draw **578k samples** from the SFT policy in our experiments (indicating that our calculation in the previous response was already overestimating the _actual_ cost of per-step DPO).
>
> ___
>
> **Theoretical justification of the cost comparison given same set of prompts:** In general, when $p$ is the accuracy of the SFT policy, the number of samples needed to draw a single correct sample is a geometric distribution with mean $1/p$. Note, that we actually need $k$, way more than a single correct sample per prompt, to ensure there is enough diversity of solutions per prompt in RFT data (from these $k$ correct ones we selected 4 correct and diverse ones, as done in Yuan et al. [68]). From tail bounds on geometric distribution, to obtain at least $k$ positive samples per prompt, if we sample $\frac{k}{p} \cdot (1+\sqrt{\log(k/\delta)})$ samples from the SFT policy, then with probability at least $1- \delta$, we will have $k$ positive traces. On the other hand, per-step DPO requires only one positive and negative sample, obtaining which requires at most $\frac{2}{p} \cdot (1+ \sqrt{\log(2/\delta)})$ many samples with probability $\geq 1 - \delta$, when $p < 0.5$. For per-step DPO, we additionally require Monte Carlo advantage estimation, which requires sampling $M$ samples per step (with at most $L$ steps in the trace). Thus, the total number of samples (from SFT policy) needed to run per-step DPO is at most $\frac{2}{p} \cdot (1+ \sqrt{\log(2/\delta)}) + M \times L$. Note that for RFT we need about $k/p$, and since we follow Yuan et al. [68], we get the best performance of RFT when $k \approx 40$ at least, which puts the $k/p$ term in RFT around **70 to 100** for MATH and GSM8k. On the other hand the $M \times L$ in per-step DPO is **at most $50$** for both of these datasets in the experiments. Thus, even in the worst-case per-step DPO (which achieves 8x scaling) requires fewer samples than the best performing variant of RFT (which achieves 2x scaling).

---

### Official Review · Reviewer_e455 · 2024-07-13

**Soundness:** 2
**Presentation:** 2
**Contribution:** 2
**Rating:** 4
**Confidence:** 3

**Summary:**

This paper investigates the impact of synthetic data on improving the reasoning capabilities of large language models (LLMs). The authors conduct an empirical study followed by a theoretical formalization to understand when and how synthetic data helps or hurts model performance on reasoning tasks.

Key findings include:-
(1) Finetuning on synthetic correct/positive problem-solution pairs offers modest gains, but sampling more correct solutions from the finetuned learner doubles sample efficiency.
(2) Training solely on model-generated positives can amplify spurious correlations, leading to flat or inverse scaling trends as data increases.
(3) Utilizing negative responses (incorrect model-generated responses) alongside positives, with appropriate credit assignment for intermediate steps, yields consistent gains over positive-only data.
(4) The proposed per-step negative data approach is equivalent to advantage-weighted reinforcement learning (RL) and helps unlearn spurious correlations.

The authors demonstrate that their per-step negative data approach improves sample efficiency by 8x compared to standard finetuning on positives. They also provide theoretical and empirical analyses to explain why credit assignment improves model generalization.

**Strengths:**

- The work reveals important nuances in using synthetic data, particularly the value of leveraging negative examples and the importance of credit assignment for intermediate steps.
- The findings offer concrete guidance for improving LLM training with synthetic data, potentially addressing data scarcity issues in language model development.
- The authors conduct extensive experiments, including scaling studies and comparisons with various baselines, strengthening the validity of their claims.

**Weaknesses:**

- While the focus on reasoning tasks is well-justified, it's unclear how well the findings generalize to other types of language tasks or domains.
- The study primarily uses 7B parameter models (DeepSeek-Math-7B and LLama2-7B). It would be valuable to see if the results hold for larger or smaller model sizes.
- The paper doesn't discuss the computational costs of their approach, which could be a significant factor in its practical application, especially for larger models or datasets.
- While the authors compare their method to several baselines, it would be beneficial to see comparisons with more recent techniques in synthetic data generation or model finetuning.
- The study doesn't address potential long-term consequences of training on synthetic data, such as potential drift or accumulation of errors over multiple generations of synthetic data.

**Questions:**

None.

**Limitations:**

Yes.

---

> ### Author Rebuttal · Authors · 2024-08-07
>
> Thank you for the feedback. To address your concerns, we clarify and add several new results to show how per-step RL on negative data can fix issues caused by accumulation of errors from training on multiple generations of positive synthetic data, and also respond to the other concerns on choice of tasks/model size, baselines, and their computational costs, and will add this discussion to the paper. **Please let us know if your concerns are addressed, and if so, we would be grateful if you are willing to raise your score.**
>
> ___
> ## [New exp.] Model collapse from training on synthetic data.
>
> **As we already discussed in the submission, our study of negative synthetic data in Sec 6 precisely addresses the concern of amplifying model biases when scaling positive synthetic data.** For this, we introduce a conceptual framework that uses negative data to identify model biases (spurious/incorrect steps) and down-weights them appropriately with step-level advantage-weighted RL. Details below:
>
> We discuss spurious correlations in Section 5, where we discuss how naively scaling data by sampling more responses for the same set of synthetic questions leads to a drop in performance (Fig 2c). We explain this finding with an experiment where we finetune on positive data with spurious steps (Fig 4). Thus, as we scale synthetic data more spurious steps creep into the training data and corrupt final performance. To further show the amplification of model biases, we add a new experiment (Fig 3a in 1-page PDF) where we train on a large amount of self-generated RFT data (128k samples) on 8k/16k questions. Note the sharp increase in test error of RFT compared to training on just 10k or 20k RFT data points (1 or 2 responses per prompt).
>
> We also add a new result in Fig 1 in the PDF to show that per-step advantage filtering of this model-generated positive and negative data allows us to address this issue of amplification of spurious steps, leading to improved test error as we scale up the number of model samples.This shows that our conceptual framework allows us to help address model biases, amplified by training on positive data alone. We will add these to the paper.
>
> ___
> ## Computational cost
>
> For a fixed size of synthetic dataset the training FLOPs for all methods SFT, RFT and all variants of DPO are identical. The main difference lies in the inference compute spent on collecting model-generated responses for RFT/DPO.
>
> For DPO, if the accuracy of $\pi_\mathrm{sft}$ is $p$, then with high probability, identifying a single positive and negative sample for a prompt takes $max(1/p, 1/(1-p))$ samples. For per-step DPO, we additionally spend $5$ samples per step to estimate advantage, for a maximum of 10 steps per question. Finally for RFT we collect positive training data, by sampling 100 prompts per question. Thus, in total, DPO requires $max(1/p, 1/(1-p))$, and  per-step DPO requires $max(1/p, 1/(1-p)) + 50$  samples per question, both of which are smaller  than $100$ for RFT, when we plug in $p \approx  0.4$ and $p \approx 0.7$ for MATH and GSM8K respectively. **Thus, per-step DPO is more computationally efficient than RFT in our experiments.**
>
> ___
> ## Comparisons with more recent techniques in synthetic data generation or model finetuning
>
> Our technique for synthetic data generation for math follows the pipeline established in Li et al. [28], Yu et al. [61], Liu et al. [30], and Wang et al. [54]. These studies already compare various ways of prompting for new questions, and thus we rely on their findings to generate synthetic problems, but we study scaling behaviors on this data. **Crucially, we also study scaling for model-generated positive data and negative data, which these prior works don’t study.**
>
> While SFT does not have many variants, we already compare w/ two different DPO baselines (Fig 5c), to underline the importance of per-step advantage computation and step-level preference pairs. We also add a new result with per-step RFT and find it improves the efficiency of RFT (Fig 1 in 1-page PDF) while addressing spurious correlations. We also add a new result showing the efficacy of per-step KTO over naive KTO alone, further strengthening our claims.
>
> If the reviewer could elaborate on specific baselines or comparisons they would like to see, we will include them in the final version.
>
> ___
> ## Justifying the choice of tasks/domains and model size used for experiments.
>
> Faced with computational constraints, at the beginning of the project, we had the choice of performing an in-depth analysis on one domain (math) or a shallow analysis over many domains. We chose the former to be able to provide detailed insight for future work on RL and synthetic data for math capabilities, which is an active area as well as to provide a roadmap for running similar analyses in other areas. Within math reasoning, our study is of a similar scope to Li et al. [28], Hwang et al. [23], etc. Finally, MATH and GSM8K datasets are the default standard for several papers. We agree that extending our analysis to other domains is interesting, and will add this as an avenue for future work.
>
> We also clarify that our conceptual framework of using positive and negative data for advantage estimation, spurious correlations in RFT, and per-step RL does not assume anything about the underlying task beyond the access to a final answer checker. Thus, our framework should be helpful for guiding algorithm design in other domains. **We also analyze how credit assignment disincentivizes memorization using a star-graph problem (Sec 6.3): while this problem is very different from the math tasks we study, we still draw conclusions that explain both of these phenomena.**
>
> **Regarding model size,** our experiments on the didactic setup uses GPT-2, so we already show that our analysis holds for smaller model sizes and reasoning tasks beyond math too (e.g., graph search). Due to computational constraints, we cannot run SFT/DPO beyond 7B.

---

> > ### Author Response · Authors · 2024-08-13
> >
> > Dear Reviewer,
> >
> > We apologize for bothering you, but since there is only one day left in the discussion period, we wanted to check in with you to see if our rebuttal addresses all outstanding concerns and have a chance to address any new ones.
> >
> > Thanks,
> > Authors

---

### Official Review · Reviewer_jvk2 · 2024-07-15

**Soundness:** 2
**Presentation:** 2
**Contribution:** 3
**Rating:** 5
**Confidence:** 4

**Summary:**

This paper studies the effect of synthetic data on improving the reasoning abilities of LLMs. The authors have compared with multiple approaches including SFT, RFT, DPO, per-step DPO, etc., and characterized the model performance w.r.t. different scales of synthetic data under each training regime. Several practical findings about improving sample efficiency are presented, including leveraging the finetuned model to sample for positive samples, utilizing per-step negative examples to alleviate the spurious correlations that might appear in the synthetic positive data, etc.

**Strengths:**

- The paper studies an important problem.
- The conceptual model of negative sample construction covered in Section 6.1 is an interesting read, as the authors formally express the synthetic data problem in the context of RL.

**Weaknesses:**

- It seems that the authors wanted to cram lots of conclusions into a single paper. While I appreciate the incentive, the paper would make a much more academic read if more experimental results are clearly presented in tables. Currently, it seems that there are few quantitative results in the paper. This also led many specific concerns (see below).

- In Lines 242-245, it reads `But, we find that even when RFT data is restricted to one solution per question, LLM trained on it outperforms SFT consistently by > 1%. Since verification is cheap, we can sample more solutions and also benefit from coverage.` Where is the evidence to this claim? In this claim, does the size of D_{syn} equal that of D_{SFT}^{+}?

- It makes me confused when I compare Lines 224-244 and Lines 245-263.

    In Lines 224-244, the authors suggest that RFT outperforms SFT in terms of sample efficiency. But in Lines 245-263, the authors suggest that RFT data contain spurious correlations, and when manually raising the percentage of spurious correlations in RFT data, SFT would outperform RFT. Especially, in Lines 248-249, `This is unlike scaling of problems and solutions in SFT data (in Figure 2(a,b)).` Since RFT might even suffer from negative scaling because of spurious correlations, I don't think that RFT could be claimed to be `more sample-efficient`.

- Also about the artificial spurious correlation amplification experiments in Lines 257-263: Does Figure 4 show the experimental results (there's no reference in the paragraph)? Does the `D_{π}^{+} spurious` correspond to the construction of `for each question in D_{syn} we sample “spurious steps” from π_{sft}`? Does this "100% spurious correlation injection" represent the authentic "spurious steps" ratio in RFT data (and what is the authentic "spurious steps" ratio in RFT data?)? Does D_{syn} contain no spurious correlation steps? As the RFT data scaling leads to negative performance, will those RFT data with a less percentage of spurious correlation steps lead to a not-so-negative performance? If this is true, can we just use the advantage defined in Eq.(3) as a stronger filtering rule and apply on RFT data? A more in-depth quantitative study is required here, so that the importance of the use of negative examples in Section 6 is better revealed.

- Is the `per-step DPO` method used in the paper proposed in the cited [23] paper? If it is true, is it also true that no new method is proposed in Section 6? Note that it is fine to focus on analysis instead of proposing new techniques, but the author should make it more clear about this in the paper. It would also be helpful if the authors briefly introduce the per-step DPO method used in the paper in a background/methodology section.

- Are the advantage in Eq.(3) and the per-step DPO method applicable to other types of reasoning tasks in addition to MATH and GSM8K? For example, ARC requires LLM reasoning (LLM often conducts this task by writing codes). Are the methods and the analysis applicable to ARC?

- Lines 343-344: `As such, we needed to tune β in Equation 1 for DPO but could not fully avoid
344 performance degradation.` Where is the evidence to this claim? Experimental results with a sweep of β would be supportive.


- Similarly, Lines 377-386 also require more experimental evidence. It would be helpful if quantitative observation is added to the claim like `When the initial advantage of a spurious step is incorrectly over-estimated, negative data algorithms up-weight the likelihood further. This only leads to further memorization. `, `leads to test-time model collapse`,  and `On the other hand, when \tilde{π} = BoK(π_{sft}) for a higher value of K, the Monte-Carlo advantage estimator has a lower variance (and error). This discussion also justifies the choice of K=5`.

- The paper lacks a Conclusion section, which I believe is because of the length limit of a submitted paper. But it is still better to add such a section and properly refactor the previous contents.


I encourage the authors to address my concerns, and I would consider raising my score provided that the issues I raised are resolved properly.

Others: Line 322: "instantation" --> "instantiation"

**Questions:**

See the Weaknesses part.

**Limitations:**

The authors have addressed the limitations.

---

> ### Author Rebuttal · Authors · 2024-08-07
>
> Thank you for the review! To address the concerns, we add many results for RFT, spurious correlations, per-step RL, and advantage estimation, which we believe improve the quality of the paper. We will use the 1 extra page in the final to incorporate them, along with a conclusion, and clarifications shown below. **Please let us know if your concerns are addressed, and if so, we would be grateful if you are willing to raise your score.**
>
> ## [New exp.] L242-245 RFT data w/ 1 sample / problem; RFT is 2x as efficient as SFT, L224-244 vs. 245-263.
>
> We compare SFT & RFT, when the RFT data is the same size as SFT (128k). For each prompt, we only include one response for RFT. On MATH we see: **44.09 (SFT) vs 45.17 (RFT)** and on GSM8K: 80.18 (SFT) to 81.27 (RFT). As said in L242-245, we observe a >1% gain.
>
> Our definition of efficiency compares performance from training on a fixed set of synthetic problems, while allowing for as many responses per problem. This is because synthetic questions are obtained by querying proprietary models, which is expensive in terms of both FLOPs and cost, and may require human verification in practice. In contrast, sampling responses from SFT policy only requires running inference against a 7B model. Please see the global response for a comparison of costs between SFT, RFT, and DPO. Fig 2 a,b shows that for an appropriate number of self-generated samples per question, RFT matches the performance of SFT with 2x more questions, hence the 2x efficiency.
>
> Of course, as we scale up RFT data, we also amplify spurious correlations as discussed in L245-263 (and per-step filtering can fix this), but for the right number of samples per prompt, **RFT does as well as SFT with 2x data**. Thus, **L224-244 & 245-263 are not contradictory**: they apply to different settings with different #samples per prompt.
> ___
>
> ## [New exp.] Per-step RFT weighted by advantage
>
> We ran an experiment with advantage filtering on all the steps present in **both** positive & negative data from the SFT policy and cloned the filtered data. This “*per-step RFT”*  outperforms standard RFT (Fig 1 in 1-page PDF), indicating that training on useful steps from negative data can improve beyond only training on positive data alone. While per-step RFT is worse than per-step DPO, we believe that this only further hints at the point that even using low advantage steps (that per-step RFT filters) for training, can further improve.
> ___
>
> ## Fig 4; Artificial spurious correlations.
>
> We will update the paper to refer to Fig. 4 in L257-263. As you noted, in these experiments, we artificially injected an arbitrary step from an incorrect response produced by SFT policy, into a positive solution trace. While this artificial injection does not necessarily reflect the proportion of spurious steps in $D_\text{syn}$, it still does provide a controlled proof-of-concept to illustrate that RFT can amplify spurious steps when present, and perform worse than SFT as a result.
> To show the presence of spurious steps in $D_\text{syn}$, we sample 128k RFT data (16 responses per prompt), and find RFT to degrade drastically (Fig 3b in 1-page PDF). For both these cases,  per-step RFT does not fall prey to spurious steps and consistently improves over SFT (Figs 1a,b, 3c in 1-page PDF).
> ___
>
> ##  [New exp.] Per-step DPO: algorithm box, comparison w/ Hwang et al.
>
> To avoid confusion, we provide an algorithm box for our per-step DPO in Panel 5 (PDF). As you note, our main focus is to analyze the importance of negative data and not to propose a new algorithm. For our experiments, we use the method from Hwang et al. [23], as this was more convenient for experimentation. That said, we have now run another version of per-step DPO (Panel 5 in 1 page PDF) derived from the advantage-weighted RL objective (Thm 6.1) and find that it improves efficiency further by 1.5-2x on MATH, supporting the efficacy of our framework.
> ___
>
> ## [New exp.] Tuning $\beta$ for DPO (L343-344)
>
> We run DPO and per-step DPO run with different $\beta$ on 128k problems (same setting as Fig 5c of the paper). Even for the best value of $\beta=0.2$ at 128k, DPO performs worse than per-step DPO and also DPO on 8k problems.
>
> | $\beta$ | DPO | per-step DPO |
> |:-------:|:--------:|:-----------------:|
> |   0.05  |   35.58  |       45.11       |
> |   0.1   |   36.18  |       47.64       |
> |   0.2   |   36.21  |       46.52       |
> |   0.3   |   35.17  |       45.89       |
> |   0.5   |   34.49  |       45.93       |
> ___
>
> ## [New exp.]  L377-386: Concrete evidence
>
> **When adv. estimation error is high…memorization is amplified**: To support this, we add a new result on the didactic star graph. We train on negative data with per-step DPO, but instead of using 5 samples for advantage estimation (as in original Fig 8), we only use 1 sample (fewer samples ⇒ larger error), and find that per-step DPO no longer learns the generalizable feature, and instead only reduces training loss by memorizing the “hard/critical’ token”, with accuracy of per-step DPO around 50%. This underscores the advantage estimation error (Fig 2a in PDF).
>
> **For a fixed sample size, advantage estimation error drops as we increase $K$.**: Based on the above observation that higher estimation error leads to memorization, we suggest computing advantages under a higher $K$. To show this, we add a new result plotting the variance of the advantages given by two independent runs of advantage estimation for each value of $K$, averaged over all steps in the SFT data (Fig 2b in PDF). As claimed, we find that $K=5$ has an estimation error that is $<25%$ of the estimation error for K=1, with larger $K$ doing better.
> ___
>
> ## Applicability to ARC.
>
> We believe that per-step RL should carry over to any task where we can break the response into steps, including code, where each line or function call can correspond to a step (e.g., prior work StepCoder (Ma et al. 23) trains process reward models for code using a similar framework).

---

> > ### Comment · Reviewer_jvk2 · 2024-08-12
> >
> > I appreciate the authors for adding the experiments. I request the authors to add these experiments into the main paper to support the claims, and also improve the presentation of the paper. I've raised the score.

---

> > > ### Author Response · Authors · 2024-08-12
> > >
> > > Thank you for increasing your score! We will definitely add in these experiments and discussion, to the main paper and improve the presentation per the discussion above.
> > >
> > > Since there is still one more day, we are also wondering if there would be some other discussion or evidence that we can provide in this period to help improve your evaluation of our paper further. Please let us know. Thanks a lot!

---

### Author Rebuttal · Authors · 2024-08-07

We thank all reviewers for their detailed feedback, and in particular would like to highlight the the positive assessment of our work by Reviewer iuJa: **“contains quite a lot of insights, and is substantially better than most papers on synthetic data that I’ve seen in the past”**.

To address the reviewers' concerns, we have added several new experiments, which we believe substantially improve the quality of our paper. The results are shown in the 1-page PDF which we will incorporate using the extra page in the final version. We also discuss a common question on the computational cost of collecting positive and negative synthetic data and running per-step RL in this global response as well. **We hope that these responses address the reviewers' concerns and look forward to the discussion period.**

___

## **List of new experiments in 1-page PDF**

- **Figure 1:** We clone responses with high advantage steps from positive and negative responses sampled from the SFT policy. We filter all responses where the minimum advantage across all steps is in the bottom 50% percentile.

- **Figure 2:** (Top) We analyze the effects of running per-step DPO on our didactic star-graph problem when advantages are computed using a single rollout and as a result are likely incorrect. (Bottom) We compute the variance of advantage estimates as we increase K for $BoK(\pi_{\mathrm{sft}})$.

- **Figure 3:** We scale up RFT to 128k data points by sampling answers from the SFT policy for only 8k/16k questions in $\mathcal{D}_\mathrm{syn}$ (left). The performance degradation for RFT is now severe, and similar to what we observe when running RFT dataset where we synthetically inject spurious steps (right). In the same figure, we also see that per-step RFT (described above) is able to filter responses with low-advantage, spurious steps and is able to almost match the performance of per-step DPO.

- **Figure 4:** We compare KTO with per-step KTO and find that the benefits of our conceptual framework of step-level credit assignment is agnostic of the choice of the underlying RL algorithm (DPO/KTO).

- **Panel 5:** We show an algorithm box for the per-step DPO algorithm used in our experiments. Note that in this algorithm we compute step-level advantages over only negative responses, consistent with Hwang et al. [23]. To more closely approximate the theoretically optimal version (mimicking advantage weighted RL) in Thm 6.1, we run a modified version of this algorithm where we also compute advantages for steps in positive rollouts. This modified version performs even better than per-step DPO validating the utility of our framework in Sec 6.

___

## **Computational costs of SFT, RFT, per-step RL**

To understand the computational costs of SFT, RFT, and per-step DPO, we make the following comparisons. First, SFT exclusively uses synthetic prompts and responses generated from more capable models. Assuming inference FLOPs are $2 \times N \times D$, this cost scales with the parameter size ($N$) of the more-capable model used to generate the data for every new synthetic question ($D$), whereas RFT and per-step DPO only require running more rollouts with a much smaller 7B model for fewer questions, not incurring the inference FLOPs of the more capable model. Thus in a FLOPs-matched comparison, RFT / per-step DPO should have an even bigger edge over SFT.

Now we perform a comparison of FLOPs / compute costs for  RFT and per-step DPO, for a fixed number of synthetic prompts. We can break the total FLOPs into two parts:
- **(a)** training FLOPs for RFT and per-step DPO.
- **(b)** inference FLOPs needed to generate data from the SFT policy for running RFT and per-step DPO, and

**Regarding (a),** we train both RFT and per-step DPO for an equal number of steps, with an identical number of forward and backward passes through the model: more precisely, since the DPO loss utilizes two samples together, we run DPO with half the batch size to fit on the GPU memory. Put together, this should lead to an equal number of forward and backward passes for per-step DPO and RFT. The training FLOPs are typically given by $6 \times N \times D$, which should be the same for both RFT and per-step DPO.

**Regarding (b),** we compare the number of samples that need to be drawn from the SFT policy for both RFT and per-step DPO. For RFT, to collect enough positives from $\pi_\mathrm{sft}$, we draw 100 samples per question and filter for positive ones. For DPO, if the accuracy of $\pi_\mathrm{sft}$ is $p$, then with high probability, identifying a single positive and negative sample for a prompt takes $ \approx(1/p, 1/(1-p))$ samples. Now, for computing advantage estimates of each step in the negative response, we set the maximum number of steps per generation as 10 and sample 5 MC rollouts conditioned on each step. Thus, in total per-step DPO requires $ \approx(1/p, 1/(1-p)) + 50$ samples per question, which is a value smaller than $100$ for RFT, when we plug in $p \approx  0.4$ and $p \approx 0.7$ for MATH and GSM8K respectively (this is the accuracy of the SFT model in our experiments). **Thus, per-step DPO is more computationally efficient than RFT in our experiments.**

---

> ### Author Response · Authors · 2024-08-10
> **Request for discussion**
>
> Dear reviewers,
>
> Apologies for bothering you! Since we are getting close to the end of the discussion period, we would be grateful and would sincerely appreciate if you could respond to our rebuttal, leaving us enough time to address any remaining questions.
>
> Thanks,
> Authors

---

### Decision · Program_Chairs · 2024-09-25

**Decision:**

Accept (poster)

**Comment:**

The paper does a thorough exploration of when synthetic data can help for training LLMs on reasoning tasks, looking at GSM8K and MATH datasets. They first report interesting general findings -- e.g. that synthetic data from an LLM helps, but what helps even more is synthetic data from fine-tuned models, and then they present the critical finding of the paper, that negative data if generated and used carefully in step-by-step settings can lead to even further improvements. Here they focus primarily on  per-step DPO, but during rebuttal they also showed that similar findings hold for KTO and RFT.
I think there are enough interesting observations and methodological contributions that make it an interesting paper, even if it is focussed very specifically on mathematical reasoning tasks. The authors reported a lot of interesting new experiments during the rebuttal period that was favorably received.